# Momentum-SAM: Sharpness Aware Minimization without Computational Overhead

**Marlon Becker**    **Frederick Altrock**    **Benjamin Risse**
University of Muenster
`{marlonbecker,f.altrock,b.risse}@uni-muenster.de`

## Abstract

The recently proposed optimization algorithm for deep neural networks Sharpness Aware Minimization (SAM) suggests perturbing parameters before gradient calculation by a gradient ascent step to guide the optimization into parameter space regions of flat loss. While significant generalization improvements and thus reduction of overfitting could be demonstrated, the computational costs are doubled due to the additionally needed gradient calculation, making SAM unfeasible in case of limited computationally capacities. Motivated by Nesterov Accelerated Gradient (NAG) we propose Momentum-SAM (MSAM), which perturbs parameters in the direction of the accumulated momentum vector to achieve low sharpness without significant computational overhead or memory demands over SGD or Adam. We evaluate MSAM in detail and reveal insights on separable mechanisms of NAG, SAM and MSAM regarding training optimization and generalization. Code is available at `https://github.com/MarlonBecker/MSAM`.

## 1   Introduction

While artificial neural networks (ANNs) are typically trained by Empirical Risk Minimization (ERM), i.e., the minimization of a predefined loss function on a finite set of training data, the actual purpose is to generalize over this dataset and fit the model to the underlying data distribution. Due to heavy overparameterization of state-of-the-art ANN models (Nakkiran *et al.*, 2021), the risk of assimilating the training data increases. As a consequence, a fundamental challenge in designing network architectures and training procedures is to ensure the objective of ERM to be an adequate proxy for learning the underlying data distribution.

One strategy to tackle this problem is to exploit the properties of the loss landscape of the parameter space on the training data. A strong link between the sharpness in this loss landscape and the models generalization capability has been proposed by Hochreiter and Schmidhuber (1994) and further analyzed in the work of Keskar *et al.* (2017). Following these works, Foret *et al.* (2021) proposed an algorithm to explicitly reduce the sharpness of loss minima and thereby improve the generalization performance, named Sharpness Aware Minimization (SAM). Built on top of gradient based optimizers such as SGD or Adam (Kingma and Ba, 2015), SAM searches for a loss maximum in a limited parameter vicinity for each optimization step and calculates the loss gradient at this ascended parameter position. To construct a computationally feasible training algorithm, SAM approximates the loss landscape linearly so that the maximization is reduced to a single gradient ascent step. Moreover, this step is performed on a single data batch rather than the full training set. Unfortunately, the ascent step requires an additional forward and backward pass of the network and therefore doubles the computational time, limiting the applications of SAM severely. Even though the linear approximation of the loss landscape poses a vast simplification and Foret *et al.* (2021) showed that searching for the maximum with multiple iterations of projected gradient ascent steps indeed yields higher maxima, these maxima, however, do not improve the generalization, suggesting that finding the actual maximum in the local vicinity is not pivotal. Instead, it appears to be sufficient

to alter the parameters to find an elevated point and perform the gradient calculation from there. Following this reasoning, the ascent step can be understood as a temporary parameter perturbation, revealing strong resemblance of the SAM algorithm to extragradient methods (Korpelevich, 1976) and Nesterov Accelerated Gradient (Nesterov, 1983; Sutskever *et al.*, 2013) which both calculate gradients at perturbed positions and were also discussed previously in the context of sharpness and generalization (Lin *et al.*, 2020a; Wen *et al.*, 2018).

Commonly, measures to address generalization issues are applied between the data distribution and the full training dataset (Keskar *et al.*, 2017; Li *et al.*, 2018). Calculating perturbations on single batches, as done by SAM, results in estimating sharpness on single batches instead of the intended full training dataset.Thus motivated, we reconsider the effect of momentum in batch-based gradient optimization algorithms as follows. The momentum vector not only represents a trace over iterations in the loss landscape and therefore accumulates the gradients at past parameter positions, but also builds an exponential moving average over gradients of successive batches. Hence, the resultant momentum vector can also be seen as an approximation of the gradient of the loss on a larger subset - in the limiting case on the full training dataset.

Building on these observations and the theoretical framework of SAM, which assumes using the entire dataset for sharpness estimations, we present Momentum-SAM (MSAM). MSAM aims to minimize the global sharpness without imposing additional forward and backward pass computations by using the momentum direction as an approximated, yet less stochastic, direction for sharpness computations. In summary, our contribution is as follows:

- We propose Momentum-SAM (MSAM), an algorithm to minimize training loss sharpness without computational overhead over base optimizers such as SGD or Adam.

- The simplicity of our algorithm and the reduced computational costs enable the usage of sharpness-aware minimization for a variety of different applications without severely compromising the generalization capabilities and performance improvements of SAM.

- We discuss similarities and differences between MSAM and Nesterov Accelerated Gradient (NAG) and reveal novel perspectives on SAM, MSAM, as well as on NAG.

- We analyze the underlying effects of MSAM, with a particular focus on the slope of the momentum vector directions, and share observations relevant for understanding momentum training beyond sharpness minimization.

- We validate MSAM on multiple benchmark datasets and neural network architectures and compare MSAM against related sharpness-aware approaches.

## 1.1 Related Work

Correlations between loss sharpness, generalization, and overfitting were studied extensively (Hochreiter and Schmidhuber, 1994; Keskar *et al.*, 2017; Lin *et al.*, 2020b; Yao *et al.*, 2018; Li *et al.*, 2018; Liu *et al.*, 2020; Damian *et al.*, 2021), all linking flatter minima to better generalization, while Dinh *et al.* (2017) showed that sharp minima can generalize too. While the above-mentioned works focus on analyzing loss sharpness, algorithms to explicitly target sharpness reduction were suggested by Zheng *et al.* (2021); Wu *et al.* (2020); Chaudhari *et al.* (2017) with SAM (Foret *et al.*, 2021) being most prevalent.

SAM relies on computing gradients at parameters distinct from the current iterations position. This resembles extragradient methods (Korpelevich, 1976) like Optimistic Mirror Descent (OMD) (Juditsky *et al.*, 2011) or Nesterov Accelerated Gradient (NAG) (Nesterov, 1983; Sutskever *et al.*, 2013) which were also applied to Deep Learning, either based on perturbations by last iterations gradients (Daskalakis *et al.*, 2018; Lin *et al.*, 2020a) or random perturbations (Wen *et al.*, 2018).

Adaptive-SAM (ASAM) (Kwon *et al.*, 2021) accommodates SAM by scaling the perturbations relative to the weights norms to take scale invariance between layers into account, resulting in a significant performance improvement over SAM. Furthermore, Kim *et al.* (2022) refine ASAM by considering Fisher information geometry of the parameter space. Also seeking to improve SAM, GSAM (Zhuang *et al.*, 2022) posit that minimizing the perturbed loss might not guarantee a flatter loss and suggest using a combination of the SAM gradient and the SGD gradients component orthogonal to the SAM gradient for the weight updates.

Unlike the aforementioned methods, several algorithms were proposed to reduce SAMs runtime, mostly sharing the idea of reducing the number of additional forward/backward passes, in contrast to our approach which relies on finding more efficient parameter perturbations. For example, Jiang *et al.*

(2023) are evaluating in each iteration if a perturbation calculation is to be performed. LookSAM (Liu *et al.*, 2022) updates perturbations only each $k$-th iterations and applies perturbation components orthogonal to SGD gradients in iterations in between. Mi *et al.* (2022) are following an approach based on sparse matrix operations and ESAM (Du *et al.*, 2022b) combines parameter sparsification with the idea to reduce the number of input samples for second forward/backward passes. Similarly, Bahri *et al.* (2021) and Ni *et al.* (2022) calculate perturbations on micro-batches. Not explicitly targeted at efficiency optimization, Mueller *et al.* (2023) show that only perturbing Batch Norm layers even further improves SAM. SAF and its memory efficient version MESA were proposed by Du *et al.* (2022a), focusing on storing past iterations weights to minimize sharpness on the digits output instead of the loss function. Perturbations in momentum direction after the momentum buffer update resulting in better performance but no speedup where proposed by Li and Giannakis (2023). Furthermore, several concepts were proposed to explain the success of SAM and related approaches going beyond sharpness reduction of the training loss landscape (Andriushchenko and Flammarion, 2022; Möllenhoff and Khan, 2023; Andriushchenko *et al.*, 2023b), with the influence on the balance of features/activations being a promising alternative explanation (Andriushchenko *et al.*, 2023a; Springer *et al.*, 2024), which we also investigate in Appx. A.6. Zhang *et al.* (2024) analyze how SAM influences adversarial robustness, a question we explore for MSAM in Appendix A.16.

## 2 Method

### 2.1 Notation

Given a finite training dataset $\mathcal{S} \subset \mathcal{X} \times \mathcal{Y}$ where $\mathcal{X}$ is the set of possible inputs and $\mathcal{Y}$ the set of possible targets drawn from a joint distribution $\mathfrak{D}$, we study a model $f_{\boldsymbol{w}} : \mathcal{X} \to \mathcal{Y}$ parameterized by $\boldsymbol{w} \in \mathcal{W}$, an element-wise loss function $l : \mathcal{W} \times \mathcal{X} \times \mathcal{Y} \to \mathbb{R}$, the distribution loss $L_{\mathfrak{D}}(\boldsymbol{w}) = \mathbb{E}_{(x,y) \sim \mathfrak{D}}(l(\boldsymbol{w}, x, y))$ and the empirical (training) loss $L_{\mathcal{S}}(\boldsymbol{w}) = 1/|\mathcal{S}| \sum_{(x,y) \in \mathcal{S}} l(\boldsymbol{w}, x, y)$. If calculated on a single batch $\mathcal{B} \subset S$ we denote the loss as $L_{\mathcal{B}}$. We denote the L2-norm by $|| \cdot ||$.

### 2.2 Sharpness Aware Minimization (SAM)

While for many datasets modern neural network architectures and empirical risk minimization algorithms, like SGD or Adam (Kingma and Ba, 2015), effectively minimize the approximation and optimization error (i.e. finding low $L_{\mathcal{S}}(\boldsymbol{w})$), reducing the generalization error ($L_{\mathfrak{D}}(\boldsymbol{w}) - L_{\mathcal{S}}(\boldsymbol{w})$) remains a major challenge. Following ideas of Hochreiter and Schmidhuber (1994), Keskar *et al.* (2017) observed a link between sharpness of the minimized empirical loss $L_{\mathcal{S}}(\boldsymbol{w}^{\text{opt}})$ with respect to the parameters and the generalization error. Intuitively, this follows from the observation that perturbations in inputs (cf. adversarial training (Goodfellow *et al.*, 2015)) and perturbations in parameters have a similar effect on network outputs (due to both being factors in matrix-vector products) and that the generalization error is caused by the limitation to a smaller input subset which resembles an input perturbation.

Without giving an explicit implementation, Keskar *et al.* (2017) sketches the idea of avoiding sharp minima by replacing the empirical loss minimization with a minimization of the highest loss value within a ball in parameter space of fixed size $\rho$:

$$\min_{\boldsymbol{w}} \max_{||\boldsymbol{\epsilon}|| \leq \rho} L_{\mathcal{S}}(\boldsymbol{w} + \boldsymbol{\epsilon}) \tag{1}$$

Foret *et al.* (2021) propose a computationally feasible algorithm to approximate this training objective via so-called Sharpness Aware Minimization (SAM). SAM heavily reduces the computational costs of the inner maximization routine of Eq. 1 by approximating the loss landscape in first order, neglecting second order derivatives resulting from the min-max objective, and performing the maximization on single batches (or per GPU in case of $m$-sharpness). These simplifications result in adding one gradient ascent step with fixed step length before the gradient calculation, i.e., reformulating the loss as

$$L_{\mathcal{B}}^{\text{SAM}}(\boldsymbol{w}) \coloneqq L_{\mathcal{B}}(\boldsymbol{w} + \boldsymbol{\epsilon}^{\text{SAM}}) \quad \text{where} \quad \boldsymbol{\epsilon}^{\text{SAM}} \coloneqq \rho \frac{\nabla L_{\mathcal{B}}(\boldsymbol{w})}{||\nabla L_{\mathcal{B}}(\boldsymbol{w})||}. \tag{2}$$

The parameters are temporarily perturbed by $\boldsymbol{\epsilon}^{\text{SAM}}$ in the direction of the locally highest slope with the perturbation removed again after gradient calculation. Thus, the parameters are not altered permanently. While performance improvements could be achieved (Foret *et al.*, 2021; Chen *et*

*al.*, 2022), the computation of $\epsilon^{\text{SAM}}$ demands an additional backward pass and the computation of $L_{\mathcal{B}}(\boldsymbol{w} + \epsilon^{\text{SAM}})$ an additional forward pass, resulting in roughly doubling the runtime of SAM compared to base optimizer like SGD or Adam.

Minimizing Eq. 2 can also be interpreted as jointly minimizing the unperturbed loss function $L_{\mathcal{B}}(\boldsymbol{w})$ and the sharpness of the loss landscape defined by

$$S_{\mathcal{B}}(\boldsymbol{w}) \coloneqq L_{\mathcal{B}}(\boldsymbol{w} + \epsilon) - L_{\mathcal{B}}(\boldsymbol{w}). \tag{3}$$

## 2.3 Momentum and Nesterov Accelerated Gradient

Commonly, SGD is used with momentum, i.e., instead of updating parameters by gradients directly ($\boldsymbol{w}_{t+1} = \boldsymbol{w}_t - \eta \nabla L_{\mathcal{B}_t}(\boldsymbol{w}_t)$ with learning rate $\eta$), an exponential moving average of past gradients is used for the updates. Given the momentum factor $\mu$ and the momentum vector $\boldsymbol{v}_{t+1} = \mu \boldsymbol{v}_t + \nabla L_{\mathcal{B}_t}(\boldsymbol{w}_t)$ the update rule becomes $\boldsymbol{w}_{t+1} = \boldsymbol{w}_t - \eta \boldsymbol{v}_{t+1}$.

The momentum vector has two averaging effects. First, it averages the gradient at different positions in the parameter space $\boldsymbol{w}_t$ and second, it averages the gradient over multiple batches $\mathcal{B}_t$ which can be interpreted as an increase of the effective batch size.

The update step consists of the momentum vector of the past iteration and the present iterations gradient. While this gradient is calculated prior to the momentum vector update in standard momentum training, NAG instead calculates the gradient after the momentum vector step is performed. The update rule for the momentum vector thus becomes $\boldsymbol{v}_{t+1} = \mu \boldsymbol{v}_t + \nabla L_{\mathcal{B}_t}(\boldsymbol{w}_t - \eta\mu\boldsymbol{v}_t)$. Analogously to Eq. 2, NAG can be formulated in terms of a perturbed loss function as

$$L_{\mathcal{B}}^{\text{NAG}}(\boldsymbol{w}) \coloneqq L_{\mathcal{B}}(\boldsymbol{w} + \epsilon^{\text{NAG}}) \quad \text{where} \quad \epsilon^{\text{NAG}} \coloneqq -\eta\mu\boldsymbol{v}_t. \tag{4}$$

Since the perturbation vector $\epsilon^{\text{NAG}}$ neither depends on the networks output nor its gradient at step $t$ no additional forward or backward pass is needed.

## 2.4 Momentum-SAM

Foret *et al.* (2021) show that performing multiple iterations of projected gradient ascent in the inner maximization does result in parameters with higher loss inside the $\rho$-ball (cf. Eq. 1). However, and counterintuitively, this improved inner maximization does not yield a better generalization of the model. We conclude that finding the exact (per batch) local maximum is not pivotal to SAM. Inspired by NAG and given that the theoretical framework of sharpness minimization is based on calculating the sharpness on the full training dataset, we propose using the momentum vector as the perturbation direction and call the resulting algorithm Momentum-SAM (MSAM) (further perturbations are discussed in Appx. A.7). Following the above notation, this yields the loss objective

$$L_{\mathcal{B}}^{\text{MSAM}}(\boldsymbol{w}) \coloneqq L_{\mathcal{B}}(\boldsymbol{w} + \epsilon^{\text{MSAM}}) \quad \text{where} \quad \epsilon^{\text{MSAM}} \coloneqq -\rho\frac{\boldsymbol{v}_t}{||\boldsymbol{v}_t||}. \tag{5}$$

Contrary to SAM, we perturb in the negative direction. While this seems counterintuitive at first glance, negative momentum directions actually cause a loss increase and are thus suitable for sharpness estimation. Since we use the momentum vector before it is updated, a step in the negative direction of the momentum has already been performed in the iteration before. We observe that this update steps overshoots the local minima in the direction of the former iterations momentum vector. Thus, when evaluated on the batch of the new iteration, the momentum direction exhibits a negative slope, caused by the high curvature in this direction. The stepsize by $\epsilon^{\text{MSAM}}$ is typically at least one order of magnitude higher than learning rate steps, so we additionally overshoot eventually occurring local minima in momentum direction and reach an increased perturbed loss which is used for sharpness minimization. We empirically validate this in detail in Sec. 4.1, Sec. A.10 and Appx. A.3.

For an efficient implementation, we shift the starting point of each iteration to be the perturbed parameters $\widetilde{\boldsymbol{w}}_t = \boldsymbol{w}_t - \rho\boldsymbol{v}_t/||\boldsymbol{v}_t||$ (in analogy to common implementations of NAG) and remove the final perturbation after the last iteration (see Alg. 1). All mentioned optimization strategies are depicted in detail in Fig. 1. Since SGD with momentum as well as Adam store a running mean of gradients, MSAM does not take up additional memory and comes with negligible computational overhead.

Furthermore, we confirm that a similar theoretical generalization bound as reported by Foret *et al.* (2021) also holds for directions of high curvature as the momentum direction (see Appx. A.1).

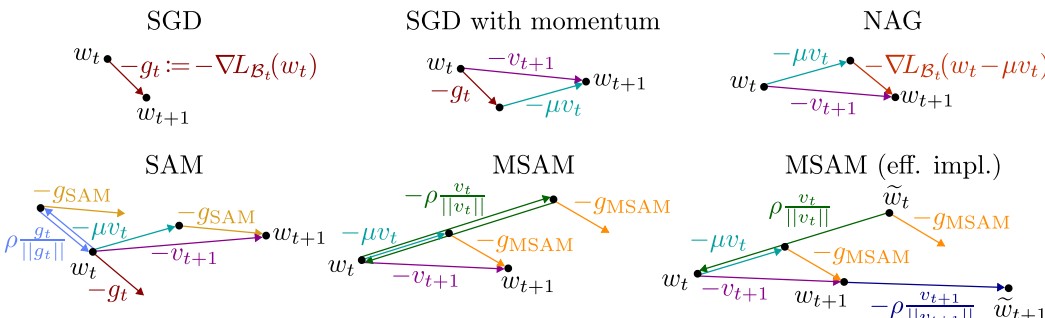

Figure 1: Schematic illustrations of optimization algorithms based on SGD. NAG calculates gradients after updating parameters with the momentum vector. SAM and MSAM calculate gradients at perturbed positions but remove perturbations again before the parameter update step. See Alg. 1 for detailed description of the efficient implementation of MSAM.

---

**Algorithm 1:** SGD with Momentum-SAM (MSAM; efficient implementation)

---

**Input:** training data $\mathcal{S}$, momentum $\mu$, learning rate $\eta$, perturbation strength $\rho$
**Initialize:** weights $\widetilde{w}_0 \leftarrow$ random, momentum vector $v_0 \leftarrow 0$
**for** $t \leftarrow 0$ **to** $T$ **do**
    sample batch $\mathcal{B}_t \subset \mathcal{S}$
    $L_{\mathcal{B}_t}(\widetilde{w}_t) = 1/|\mathcal{B}_t| \sum_{(x,y)\in\mathcal{B}_t} l(\widetilde{w}_t, x, y)$
    $g_{\text{MSAM}} = \nabla L_{\mathcal{B}_t}(\widetilde{w}_t)$                       // inc pert.
    $w_t = \widetilde{w}_t + \rho\frac{v_t}{||v_t||}$               // remove last pert.
    $v_{t+1} = \mu v_t + g_{\text{MSAM}}$               // update momentum
    $w_{t+1} = w_t - \eta v_{t+1}$                    // SGD step
    $\widetilde{w}_{t+1} = w_{t+1} - \rho\frac{v_{t+1}}{||v_{t+1}||}$         // next pert.
**end**
$w_T = \widetilde{w}_T + \rho\frac{v_T}{||v_T||}$                        // remove pert.
**return** $w_T$

---

## 3 Experimental Results

### 3.1 Speed and Accuracy for ResNets on CIFAR100

In Tab. 1, we show test accuracies for MSAM and related optimizers for WideResNet-28-10, WideResNet-16-4 (Zagoruyko and Komodakis, 2016) and ResNet50 (He *et al.*, 2016) on CIFAR100 (Krizhevsky and Hinton, 2009) and vision transformers (Dosovitskiy *et al.*, 2021) and ImageNet-1k (Deng *et al.*, 2009) next to the training speed. We first tuned the learning rate and weight decay for SGD/AdamW and then optimized $\rho$ for each model (see Appx. A.13 for more details). Additionally,

Table 1: Comparison against multiple (sharpness-aware) optimizers. Baseline optimizers are SGD for CIFAR100 and AdamW for ImageNet. Please see Appx. A.14 for experimental details. MSAM outperforms optimizers of equal speed (AdamW/SGD and NAG) and alternative approaches for faster sharpness reduction.

| Optimizer | CIFAR100 | | | ImageNet | Speed |
| --- | --- | --- | --- | --- | --- |
| | WRN-28-10 | WRN-16-4 | ResNet-50 | ViT-S/32 | |
| SAM | $84.16_{\pm 0.12}$ | $79.25_{\pm 0.10}$ | $83.36_{\pm 0.17}$ | 69.1 | 0.52 |
| Baseline | $81.51_{\pm 0.09}$ | $76.90_{\pm 0.15}$ | $81.46_{\pm 0.13}$ | 67.0 | **1.00** |
| NAG | $82.00_{\pm 0.11}$ | $77.09_{\pm 0.18}$ | $82.12_{\pm 0.12}$ | – | **0.99** |
| LookSAM | $\mathbf{83.31_{\pm 0.12}}$ | $\mathbf{79.00_{\pm 0.08}}$ | $82.24_{\pm 0.11}$ | 68.0 | 0.84 |
| ESAM | $82.71_{\pm 0.38}$ | $77.79_{\pm 0.11}$ | $80.49_{\pm 0.40}$ | 66.1 | 0.62 |
| MESA | $82.75_{\pm 0.08}$ | $78.32_{\pm 0.08}$ | $81.94_{\pm 0.26}$ | **69.0** | 0.77 |
| MSAM (ours) | $\mathbf{83.21_{\pm 0.07}}$ | $\mathbf{79.11_{\pm 0.09}}$ | $82.65_{\pm 0.12}$ | **69.1** | **0.99** |

we conducted experiments with related approaches which seek to make SAM more efficient, namely ESAM (Du *et al.*, 2022b), LookSAM (Liu *et al.*, 2022) and MESA (Du *et al.*, 2022a). While Du *et al.* (2022a) also proposed a second optimizer (SAF), we decided to compare against the memory-efficient version MESA (as recommended by the authors for e.g. ImageNet). Note that LookSAM required tuning of an additional hyperparameter. See Appx. A.14 for implementation details on the related optimizers. Optimizers of the same speed as MSAM (i.e. SGD/AdamW and NAG) are significantly outperformed. While SAM reaches slightly higher accuracies than MSAM, twice as much runtime is needed. Accuracies of MSAM and LookSAM do not differ significantly for WideResNets, however, MSAM performs better on ResNet-50, is faster, and does not demand additional hyperparameter tuning. For ESAM we observed only a minor speedup compared to SAM and the accuracies of MSAM could not be reached. MESA yields similar results to MSAM for ViT on ImageNet but performs worse on all models on CIFAR100 and is slower compared to MSAM.

## 3.2 ResNet and ViT on ImageNet Results

Moreover, we test MSAM for ResNets (He *et al.*, 2016) and further ViT variants (Dosovitskiy *et al.*, 2021) on ImageNet-1k (Deng *et al.*, 2009) and report results in Tab. 2. Due to limited computational resources, we only run single iterations, but provide an estimate of the uncertainty by running 5 iterations of baseline optimizers for the smallest models per category and calculate the standard deviations. During the learning rate warm-up phase commonly used for ViTs we set $\rho_{\mathrm{MSAM}} = 0$. SAM also benefits from this effect, but less pronounced, so we kept SAM active during warm-up phase to stay consistent with related work (see Appx. A.12 for detailed discussion). While performance improvements are small for ResNets for MSAM and SAM, both optimizers achieve clear improvements for ViTs. Even though slightly below SAMs performance for most models, MSAM yields comparable results while being almost twice as fast.

In addition, we conducted experiments for ViT-S/32 on ImageNet when giving MSAM the same computational budget as SAM (i.e. training for 180 epochs) yielding a test accuracy of 70.1% and thus clearly outperforming SAMs 69.1% (also see Appx. A.9).

Table 2: Test accuracies on ImageNet for baseline optimizers (SGD or AdamW), SAM and MSAM. Estimated uncertainties: ResNet: $\pm 0.08$, ViT (90 epochs): $\pm 0.17$, ViT (300 epochs): $\pm 0.24$. Improvements over baseline are given in green. MSAM yields results comparable to SAM for most models while being $\approx 2$ times faster in all our experiments.

| Model | Epochs | Baseline | | SAM | MSAM |
|---|---|---|---|---|---|
| ResNet-50 | 100 | SGD | 76.3 | $76.6_{+0.3}$ | $76.5_{+0.2}$ |
| ResNet-101 | 100 | SGD | 77.9 | $78.7_{+0.8}$ | $78.2_{+0.3}$ |
| ViT-S/32 | 300 | AdamW | 67.2 | $71.4_{+4.2}$ | $70.5_{+3.3}$ |
| | 90 | AdamW | 67.0 | $69.1_{+2.1}$ | $69.1_{+2.1}$ |
| ViT-S/16 | 300 | AdamW | 73.0 | $78.2_{+5.2}$ | $75.8_{+2.8}$ |
| | 90 | AdamW | 72.6 | $75.8_{+3.2}$ | $74.9_{+2.3}$ |
| ViT-B/32 | 90 | AdamW | 66.9 | $70.4_{+3.5}$ | $70.2_{+3.3}$ |
| ViT-B/16 | 90 | AdamW | 73.0 | $77.7_{+4.7}$ | $75.7_{+2.7}$ |

## 3.3 Combination with other SAM Variants

As shown by Kwon *et al.* (2021), weighting the perturbation components by the parameters significantly improves SAM. Similarly, Mueller *et al.* (2023) showed that applying the perturbations only to the Batch Norm layers (Ioffe and Szegedy, 2015) yields further enhancements. Both of these techniques can also be applied to MSAM, yielding test results similar to SAM (see Tab. 3).

## 4 Properties of MSAM

In the following we analyze the perturbation direction of MSAM, its similarities to SAM and the loss sharpness in more detail. Additional experiments are presented in the appendix. These include the observations that the perturbation normalization is not pivotal to MSAM and instead the binding to the learning rate $\eta$ discriminates NAG and MSAM (Appx. A.2) and that MSAM reduces feature ranks more effectively than SAM (Appx. A.6).

Table 3: Test accuracy for different variants of MSAM/SAM on CIFAR100. Adaptive refers to ASAM (Kwon *et al.*, 2021) and BN-only to applying the perturbance only to Batch Norm layers (cf. Mueller *et al.* (2023)). MSAM performs well with both variants.

| Optimizer | | WRN-28-10 | WRN-16-4 | ResNet-50 |
|---|---|---|---|---|
| SGD | | $81.51_{\pm 0.09}$ | $76.90_{\pm 0.15}$ | $81.46_{\pm 0.13}$ |
| vanilla | SAM | $84.16_{\pm 0.12}$ | $79.25_{\pm 0.10}$ | $83.36_{\pm 0.17}$ |
| | MSAM | $83.21_{\pm 0.07}$ | $79.11_{\pm 0.09}$ | $82.65_{\pm 0.12}$ |
| adaptive | SAM | $84.74_{\pm 0.13}$ | $79.96_{\pm 0.13}$ | $83.30_{\pm 0.06}$ |
| | MSAM | $84.15_{\pm 0.13}$ | $79.89_{\pm 0.09}$ | $83.48_{\pm 0.08}$ |
| BN-only | SAM | $84.57_{\pm 0.07}$ | $79.73_{\pm 0.24}$ | $84.51_{\pm 0.17}$ |
| | MSAM | $83.62_{\pm 0.09}$ | $79.73_{\pm 0.14}$ | $83.49_{\pm 0.19}$ |

## 4.1 Negative and Positive Perturbation Directions

Instead of ascending along the positive gradient as in SAM, we propose perturbing along the negative momentum vector (positive $\rho^{\text{MSAM}}$ in our notation) as it is also done by extragradient methods as by Lin *et al.* (2020a). Counterintuitively, the cosine similarity between the momentum vector $v_{t-1}$ and the gradient $g_t = \nabla L_{\mathcal{B}_t}(w_t)$ is negative. Thus, the negative momentum direction actually has a positive slope, so that perturbing in this direction resembles an ascent on the per-batch loss. An update step in the momentum direction was already performed and the momentum direction is typically of high curvature (see Appx. A.4). The positive slope of the negative momentum direction can thus be explained by the parameter update overshooting local minima if evaluated on the new iterations batch. As a consequence, moving further in this direction yields valid sharpness estimates. If not already caused by the SGD update step, the overshooting perturbation step will overshoot the minima since the used perturbations are typically at least one order of magnitude larger than the optimization step sizes. Positive momentum perturbations, instead, yield improper estimates of the local sharpness since the minima which was overshot is approached again. We analyze this effect in detail in Appx. A.3. In addition, we provide direct evidence that the proposed perturbation results in a loss increase suitable for sharpness estimations in Appx. A.10. The increased generalization of MSAM directly follows from this phenomenon, which we show in our theoretical consideration in Appx. A.1 based on a PAC-Bayes bound.

We did not observe any increase in test performance when perturbing in the positive momentum

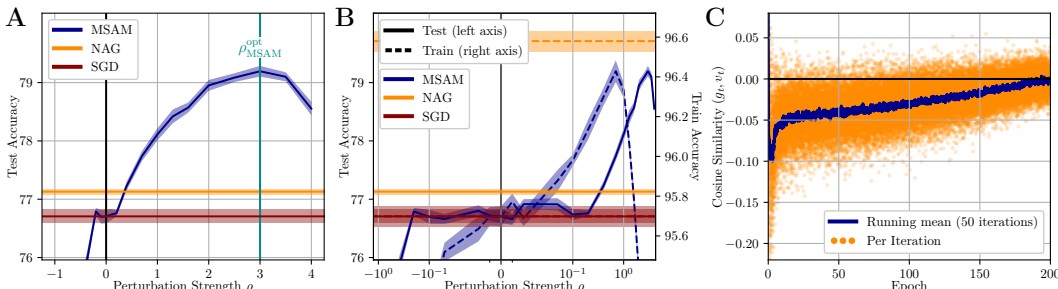

Figure 2: WideRestNet-16-4 on CIFAR100 **A**: Test accuracy for positive and negative $\rho$ compared against SGD and NAG. **B**: Train and test accuracy on logarithmic scale. **C**: Cosine similarity between momentum vector $v_{t-1}$ and gradient $g_t = \nabla L_{\mathcal{B}_t}(w_t)$. Momentum vector direction has mostly negative slope during training and approaches zero at the end (caused by cosine learning rate scheduler).

direction (negative $\rho^{\text{MSAM}}$). In fact, negative $\rho$ values cause a rapid decrease in test accuracy, whereas positive $\rho$ values cause a gain in test accuracy of more than $2\%$ for a WideResNet-16-4 trained on CIFAR100 as depicted in Fig. 2B, while NAG only provides minor improvements.

Fig. 2C shows the same data on logarithmic scale next to the test accuracy. The ordinate limits are chosen such that baseline (SGD) accuracies as well as maximal gains by MSAM align. NAG improves the training accuracy greatly, especially compared to the gains in test accuracy. This underlines that NAG is designed to foster the optimization procedure (ERM) but does not improve the generalization capabilities of the model. Similarly, for MSAM the maximal test accuracy is reached for high values of $\rho$ where the train accuracy dropped far below the baseline, emphasizing the effect of MSAM on the generalization instead of optimization.

Furthermore, small negative values of $\rho$ induce a steep decrease in training accuracy while the test accuracy is not significantly affected, but drops for higher negative $\rho$. So the generalization improves also for negative $\rho$ values, however, the test performance does not, since the improved generalization is overlaid by the decreasing training performance. This offers an additional explanation why MSAM does not improve the test accuracy for positive momentum perturbations. A perturbation in the positive momentum vector direction resembles a step back to past iterations which might result in the gradient not encoding appropriate present or future information about the optimization direction and thus seems to be ill-suited to reduce the training loss effectively, counteracting the benefits of the increased generalization (smaller test-train gap). SAM might not suffer from this effect, since the local (per batch) gradient does not encode much of the general optimization direction (which is dominated by the momentum vector), hence, the perturbed parameters disagree with parameters from previous iterations.

## 4.2 Similarity between SAM and MSAM

To support our hypotheses that MSAM yields a valid approximation for SAM's gradient calculations, we investigate the similarity between the resulting gradients. After searching for the optimal value, we keep $\rho_{\text{SAM}} = 0.3$ fixed, train a model with SAM, and calculate the gradients

$$\boldsymbol{g}_{\text{SGD}} = \nabla L_{\mathcal{B}_t}(\boldsymbol{w}_t)$$
$$\boldsymbol{g}_{\text{SAM}} = \nabla L_{\mathcal{B}_t}(\boldsymbol{w}_t + \rho_{\text{SAM}}\nabla L_{\mathcal{B}_t}(\boldsymbol{w}_t)/||\nabla L_{\mathcal{B}_t}(\boldsymbol{w}_t)||)$$
$$\boldsymbol{g}_{\text{MSAM}}(\rho_{\text{MSAM}}) = \nabla L_{\mathcal{B}_t}(\boldsymbol{w}_t - \rho_{\text{MSAM}}\boldsymbol{v}_t/||\boldsymbol{v}_t||)$$

while we keep $\rho_{\text{MSAM}}$ as a free parameter. To eliminate gradient directions which do not contribute in distinguishing between SAM and SGD gradients, we first project $\boldsymbol{g}_{\text{MSAM}}$ into the plane spanned by $\boldsymbol{g}_{\text{SAM}}$ and $\boldsymbol{g}_{\text{SGD}}$ and then calculate the angle $\theta$ to $\boldsymbol{g}_{\text{SAM}}$ (see Fig. 3A). By repeating this every 50 iterations for various values $\rho_{\text{MSAM}}$ and calculating the value of zero-crossing $\rho_0$, we determine when the maximal resemblance to SAM is reached (see Fig. 3B). As shown in Fig. 3C, $\rho_0$ reaches values close to the optimal regime of $\rho_{\text{MSAM}}^{\text{opt}} \approx 3$ (cf. Fig. 2A) for most epochs. While this correlation does not yield strict evidence it offers additional empirical support for the similarity between SAM and MSAM gradients.

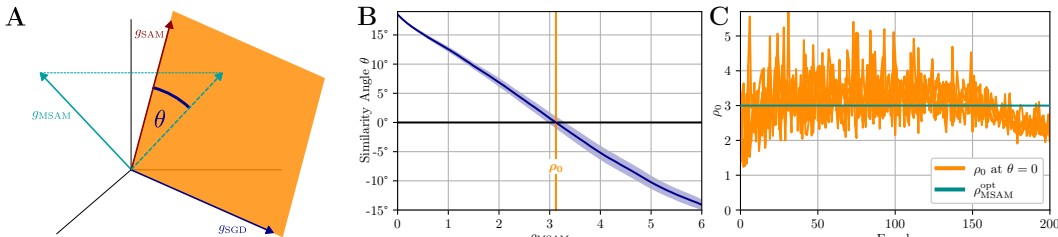

Figure 3: **A**: Projecting $g_{\text{MSAM}}$ into the plane of $g_{\text{SAM}}$ and $g_{\text{SGD}}$ to measure SAM/MSAM similarity. **B**: Varying $\rho_{\text{MSAM}}$ until maximal similarity is reached (i.e. $\theta = 0$) and determine $\rho_0$. **C**: $\rho_{\text{MSAM}}$ at maximal similarity $\rho_0$ is close to generalization optimality ($\rho_{\text{MSAM}}^{\text{opt}}$, cf. Fig. 2A) for most epochs.

## 4.3 Loss Sharpness Analysis

As mentioned above, the SAM implementation performs the loss maximization step on a single data batch instead of the full training set ($L_{\mathcal{B}}$ vs. $L_{\mathcal{S}}$). To analyze the efficacy of SAMs sharpness minimization, we therefore compare the sharpness (cf. Eq. 3) in the direction of local (per batch) gradients for models after full training with SGD, SAM and MSAM as a function of the perturbation scale $\rho$ in Fig. 4A. The minima in local gradient directions are shifted from $\rho = 0$, since parameters found after training are usually not minima but saddle points (Dauphin *et al.*, 2014). Compared to the other optimizers, SAM successfully minimizes the sharpness, especially at optimal $\rho_{\text{opt}}$ (as used during training).

The sharpness in momentum direction (Fig. 4B) represents the MSAM objective (Eq. 5). Here we do not include the negative sign in the definition of $\boldsymbol{\epsilon}$ (as in Eq. 5), hence $\rho_{\text{opt}}$ is negative. As expected, MSAM reduces this sharpness best. In contrast to the definition before, the sharpness is

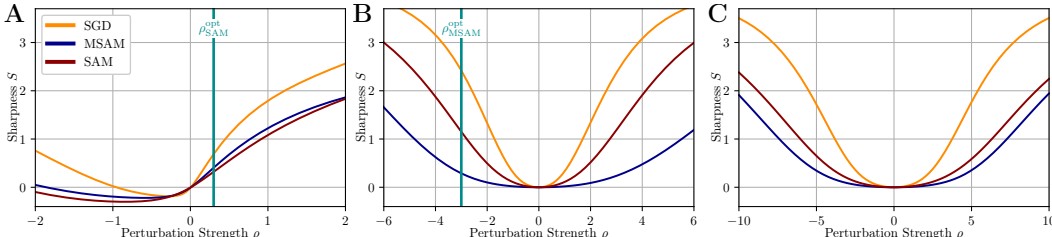

Figure 4: Sharpness (Eq. 3) after full training (200 epochs) for WideResNet-16-4 on CIFAR100 along different directions $\epsilon$ scaled by $\rho$. **A**: Gradient direction (as used as perturbation in SAM). **B**: Momentum direction (as in MSAM). **C**: Random filter-normalized direction as in Li *et al.* (2018). Vertical line at $\rho^{\text{opt}}$ marks values of optimal test performance (cf. Fig. A.15A). MSAM and SAM are reducing their respective optimization objective best while MSAM reaches the lowest sharpness along random directions.

symmetric now for positive and negative signs. While MSAM only minimizes the sharpness in the negative direction explicitly, the positive direction is reduced jointly, further supporting the validity of perturbations in the negative momentum direction.

In Fig. 4C, we choose filter-normalized random vectors as perturbations as in Li *et al.* (2018). Since the loss landscape is rotational symmetric around the origin for multiple directions (as used in the original paper), we confine our analysis to one perturbation direction allowing an easier comparison (see Appx. A.11 for two perturbation directions). MSAM reaches the lowest sharpness, while both MSAM and SAM, significantly flatten the loss. This might be caused by MSAM approximating the maximization of $L_S$ better due to the momentum vector $v_t$ being an aggregation of gradients of multiple batches. Interestingly, this relates to the findings of Foret *et al.* (2021) regarding $m$-sharpness, where the authors performed the maximization on even smaller data samples (fractions of batches per GPU in distributed training), yielding even better generalization. In this sense MSAM, reduces $m$ even further over ordinary SAM ($m = 1$). In the same line of argument and despite being more efficient, MSAM oftentimes does not improve generalization compared to SAM. However, contradicting the general idea behind correlations of generalization and sharpness, MSAM yields flatter minima (if defined as in Fig. 4C), hence, indicating that explanations for the improved generalization of SAM/MSAM go beyond the reduction in sharpness. Additionally, we show that SAM guides the optimization through regions of lowest curvature compared to SGD and SAM in Appx. A.4 and yields low Hessian eigenvalues Appx. A.5.

## 5    Conclusion

In this work we introduced Momentum-SAM (MSAM), an optimizer achieving comparable results to the SAM optimizer while requiring no significant computational or memory overhead over optimizers such as Adam or SGD, hence, halving the computational load compared to SAM and thus reducing the major hindrance for a widespread application of SAM-like algorithms when training resources are limited. Alternative efficient approaches (which are mostly based on simple sparsifications) are all outperformed in speed and accuracy. Instead, we showed that perturbations independent of local gradients can yield significant performance enhancements. In particular perturbations in the negative momentum buffer direction yield substantial generalization improvements. While the negative sign of the perturbations seemed counterintuitive at a first glance, we analyzed the loss landscape in momentum direction in detail to show that a negative perturbation actually causes a loss ascent (please also see the appendix for further analysis).

In summary, we not only proposed a new optimization method, but also offered a detailed empirical analysis yielding multiple new perspectives on sharpness aware optimization and momentum training in general.

## Acknowledgements

This work was funded by the German Research Foundation (DFG CRC 1459 Intelligent Matter - Project-ID 433682494).

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

## Appendix

### A.1 Theoretical Analysis of MSAM

We build our analysis in analogy to Foret *et al.* (2021). While Foret *et al.* (2021) proofed the existence of an upper generalization bound if the parameters with the highest loss in a fixed $\rho$-ball are found, we show that a similar bound can also be derived by simply assuming perturbations in directions of high curvature. In practice this first assumption is fulfilled by the momentum vector $\boldsymbol{v}_t$, since the directions of gradients are directions of high curvature. Secondly, if a properly tuned learning rate is used, the slope in momentum direction after the parameter update is either close to zero or even negative caused by overshooting marginally.
We state these two assumptions in Setting 1. While we empirically validate the first assumption in Appx. A.4, we already showed and discussed evidence for the second assumption in the main text (cf. Fig. 2C).

**Proposition 1** *Let $\boldsymbol{\epsilon}, \boldsymbol{v} \in \mathcal{W}$ with i.i.d. components $\epsilon_i \sim \mathcal{N}(0, \sigma)$ for some $\sigma > 0$, then for any $\rho > 0$*

$$\mathbb{E}[\mathbf{1}_{\{\|\boldsymbol{\epsilon}\|_2 \leq \rho\}} \boldsymbol{\epsilon}^T \text{Hess}(L_{\mathcal{S}}(\boldsymbol{w})) \boldsymbol{\epsilon}] \leq \rho^2 \kappa, \tag{A.1}$$

*where $\kappa := \frac{1}{|\mathcal{W}|} \text{tr}[\text{Hess}(L_{\mathcal{S}}(\boldsymbol{w}))]$.*

**Proof** *W.L.O.G. we assume $\text{Hess}(L_{\mathcal{S}}(\boldsymbol{w}))$ to be diagonal. The claim then follows from the linearity of the expectation and symmetry.* $\square$

**Setting 1** *Let $\boldsymbol{w}, \boldsymbol{v} \in \mathcal{W}$ with $\|\boldsymbol{v}\|_2 = 1$ such that:*

- $\boldsymbol{v}^T \text{Hess}(L_{\mathcal{S}}(\boldsymbol{w})) \boldsymbol{v} > \kappa$,

- $\nabla L_S(\boldsymbol{w}) \cdot \boldsymbol{v} \leq 0$.

**Lemma 2** *Assume Setting 1 and let $\boldsymbol{\epsilon} \in \mathcal{W}$ with $\epsilon_i \sim \mathcal{N}(0, \sigma)$, then it holds for any $\rho > 0$ that*

$$\mathbb{E}[\mathbf{1}_{\{\|\boldsymbol{\epsilon}\|_2 \leq \rho\}} L_{\mathcal{S}}(\boldsymbol{w} + \boldsymbol{\epsilon})] \leq L_{\mathcal{S}}(\boldsymbol{w} - \rho \boldsymbol{v}) + \mathcal{O}(\rho^3).$$

**Proof** *Applying a Taylor expansion around $\boldsymbol{w}$ yields:*

$$\mathbb{E}[\mathbf{1}_{\{\|\boldsymbol{\epsilon}\|_2 \leq \rho\}} L_{\mathcal{S}}(\boldsymbol{w} + \boldsymbol{\epsilon})] \leq L_{\mathcal{S}}(\boldsymbol{w} - \rho \boldsymbol{v})$$

$$\Longleftrightarrow \underbrace{\mathbb{E}[\mathbf{1}_{\{\|\boldsymbol{\epsilon}\|_2 \leq \rho\}} \nabla L_S(\boldsymbol{w}) \cdot \boldsymbol{\epsilon}]}_{=0} + \underbrace{\mathbb{E}[\mathbf{1}_{\{\|\boldsymbol{\epsilon}\|_2 \leq \rho\}} \boldsymbol{\epsilon}^T \text{Hess}(L_{\mathcal{S}}(\boldsymbol{w})) \boldsymbol{\epsilon}]}_{\leq \rho^2 \kappa} \leq$$

$$\underbrace{-\rho \nabla L_S(\boldsymbol{w}) \cdot \boldsymbol{v}}_{\geq 0} + \rho^2 \boldsymbol{v}^T \text{Hess}(L_{\mathcal{S}}(\boldsymbol{w})) \boldsymbol{v} + \mathcal{O}(\rho^3)$$

$$\Longleftarrow \kappa \leq \boldsymbol{v}^T \text{Hess}(L_{\mathcal{S}}(\boldsymbol{w})) \boldsymbol{v} + \mathcal{O}(\rho^3)$$

*subtracting the $\mathcal{O}(\rho^3)$-term from the initial inequality then yields the claim.* $\square$

**Theorem 3** *Assume Setting 1 then for any distribution $\mathfrak{D}$, with probability $1 - \delta$ over the choice of the training Set $\mathcal{S} \sim \mathfrak{D}$,*

$$L_{\mathfrak{D}}(\boldsymbol{w}) \leq L_{\mathcal{S}}(\boldsymbol{w} - \rho \boldsymbol{v}) + \sqrt{\frac{\dim(\mathcal{W}) \log\left(1 + \frac{\|\boldsymbol{w}\|_2^2}{\rho^2}\left(1 + \sqrt{\frac{\log(|\mathcal{S}|)}{\dim(\mathcal{W})}}\right)^2\right) + 4 \log \frac{|\mathcal{S}|}{\delta} + \mathcal{O}(1)}{|\mathcal{S}| - 1}} + \mathcal{O}(\rho^3)$$

**Proof** *Using the bound from Lemma 2 we adapt the proof of Theorem 2 in Foret* et al. *(2021) (i.e. Eq. 13 and following) to show the claim.* $\square$

### A.2 Normalization

To gain a better understanding of the relation between MSAM and NAG, we conducted an ablation study by gradually altering the MSAM algorithm until it matches NAG. Firstly, we drop the

normalization of the perturbation $\epsilon$ (numerator in Eq. 5), then we reintroduce the learning rate $\eta$ to scale $\epsilon$, and finally set $\rho = 1$ to arrive at NAG. Train and test accuracies as functions of $\rho$ are shown in Fig. A.1 for all variants. Dropping the normalization only causes a shift of $\rho$ indicating that changes of the momentum norm during training are negligible (contradicting Dai *et al.* (2023)). However, scaling by $\eta$ drastically impacts the performance. Since the model is trained with a cosine learning rate scheduler (Loshchilov and Hutter, 2017), $\rho$ decays jointly with $\eta$. The train accuracy is increased significantly not only for $\rho = 1$ (NAG), but even further for higher $\rho$, while the test performance drops at the same time when compared to MSAM. Thus, optimization is improved again while generalization is not, revealing separable mechanisms for test performance improvements of MSAM and NAG. High disturbances compared to the step size at the end of training appear to be crucial for increased generalization. Extensively investigating the effect of SAM/MSAM during different stages of training might offer potential to make SAM even more effective and/or efficient (i.e. by scheduling $\rho$ to only apply disturbances for selected episodes).

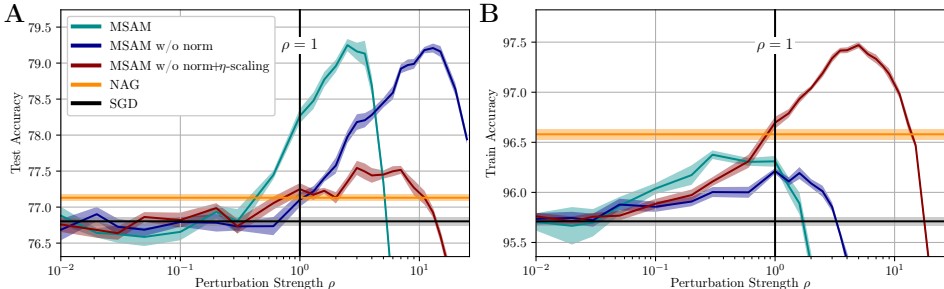

Figure A.1: Test (**A**) and train (**B**) accuracy for WideResNet-16-4 on CIFAR100 for different normalization schemes of MSAM in dependence on $\rho$. MSAM without normalization works equally well. If the perturbation $\epsilon$ is scaled by learning rate $\eta$ train performance (optimization) is increased while test performance (generalization) benefits only marginally.

## A.3    Detailed Analysis of Negative Perturbation Direction

### A.3.1    Overshooting of Local Minima

As discussed in the main text, negative momentum perturbations lead to a loss increase, which in turn results in effective sharpness minimization. This effect is caused by overshooting local minima in the momentum direction during the SGD update. We validate this by depicting the 1D-loss landscape along the momentum direction $v_{t-1}$ (before being updated) calculated on the next iterations batch in Fig. A.2. As can be seen the SGD update from $w_{t-1}$ to $w_t$ does not approach the local minima on the new batch but instead overshoots and results in a negative slope of the momentum buffer direction (i.e. negative cosine similarity to the gradient). Thus, further perturbations (dashed arrow) in the negative momentum direction result in an increase of the loss, allowing for effective sharpness estimations. It is important to note that the loss is calculated on the new unused batch and the momentum buffer is not updated yet. After updating the momentum buffer with the new batches gradient to $v_t$, the slope of the new momentum direction turns positive again allowing for SGD to behave properly (cf. Appx. A.3.2). Additional analyses validating that overshooting is indeed the cause of negative momentum slopes are given in Appx. A.3.2.

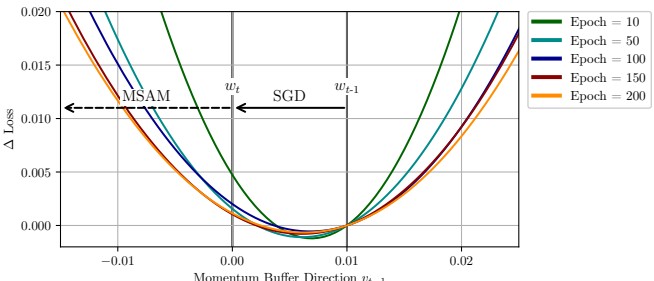

Figure A.2: Loss change (i.e. sharpness) in momentum buffer direction $v_{t-1}$ during optimization. WRN-16-4 on CIFAR100 averaged over 30 runs. Minima are overshot resulting in loss increase for further perturbations in negative momentum direction as done by MSAM. Effect even occurs for low (constant) learning rate of $\eta = 0.01$ used here while we use $\eta = 0.5$ in the main manuscript.

We measure the effect of by the negative cosine similarity between the momentum vector $v_{t-1}$ (before being updated) and the gradients. Additionally, to Fig. 2C we make similar observations for all of our studied models and datasets in Fig. A.3.

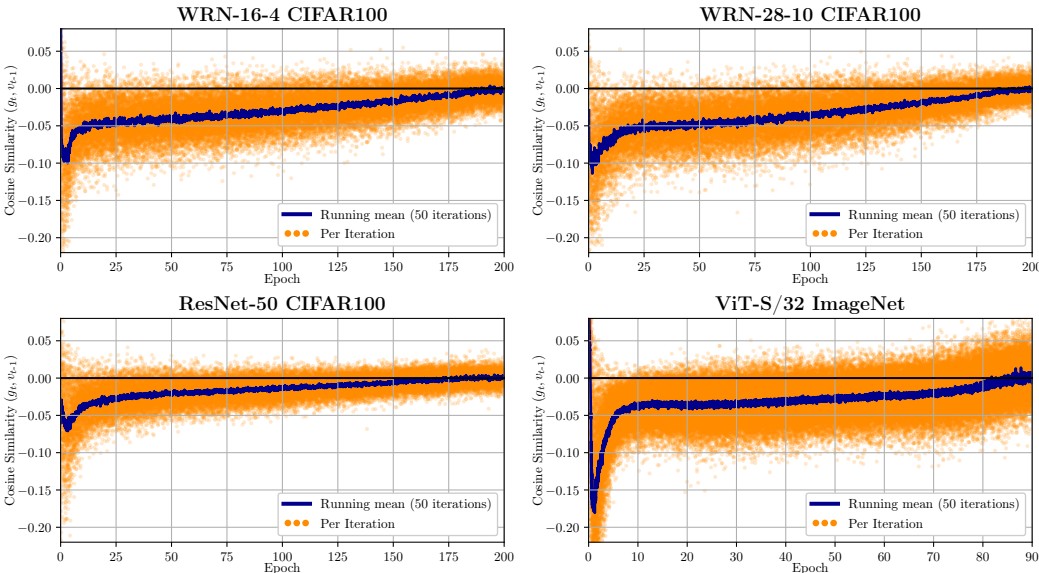

Figure A.3: Cosine similarity between momentum vector $\boldsymbol{v}_{t-1}$ and gradient $g_t = \nabla L_{\mathcal{B}_t}(\boldsymbol{w}_t)$ for different models/datsets. all with cossine scheduler

### A.3.2 Overshooting as Cause of Negative Momentum Slope

To validate that the effect of negative slope of the momentum direction (i.e. negative cosine similarity to the gradient) is caused by the overshooting of minima, we perform the following experiments depicted in Fig. A.4.

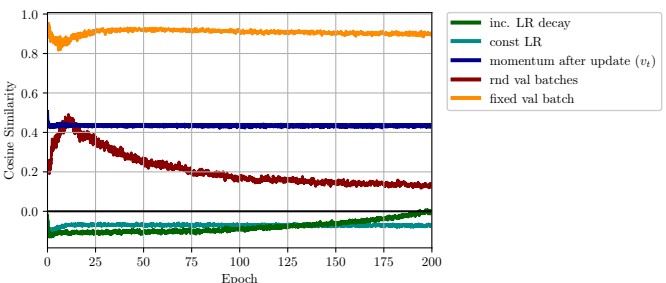

Figure A.4: Cosine similarity between momentum and gradient with and without LR decay (cosine scheduler), after performing the momentum update with the new batches gradient, for random and a single fixed batch sampled from the validation dataset. WRN-16-4 on CIFAR100. Running average over 50 iterations.

Only when using a learning rate scheduler, the cosine similarity approaches zero to the end of the training. Since the learning rate decays close to zero at the end of the training, minima are not overshot anymore.

After updating the momentum buffer with the current gradient, the slope turns positive with the cosine similarity to the gradient being slightly below 0.5 since the updated momentum buffer $v_t$ is a composition of the gradient itself (weighted by $\mu = 0.9$) and the former momentum buffer. This allows for an effective Empirical Risk Minimization (ERM) by SGD despite the overshooting of minima along the not-updated momentum direction.

For the last two experiments, we sample batches from the validation dataset, which are not used for parameter updates, and thus yield gradients independent of the directions used for optimization. We use these to construct an additional pair of gradients and a moving average of gradients (as the momentum buffer) and calculate the cosine similarity between these two. Note that the validation batches are sampled additionally to the training samples and are not used in any parameter updates. When sampling only a single fixed batch from the validation set for gradient calculation as well as construction of the moving average, the cosine similarity is close to 1 (fully correlated). Thus, the parameter updates of the loss landscape (which are not correlated to the sampled batch), only have minor impact on the gradient change.

However, if new random additional batches are sampled from the validation dataset in each iteration, the cosine similarity stays positive, since no parameter updates are performed in these directions and thus no overshooting occurs. In the first epochs the correlation the of the moving average and gradients of new sampled batches even increases. We hypothesize that the common optimization direction, when general information is learned in the early stages, causes this effect. In the later stages of training, however, gradient information becomes more sample-specific.

### A.3.3 Dependence of Cosine Similarity on Learning Rate $\eta$, Momentum Factor $\mu$ and Batch Size

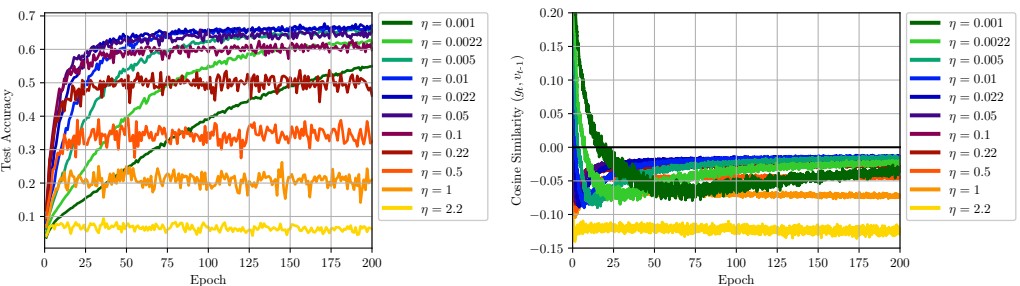

Figure A.5: WRN-16-4 on CIFAR100. Constant learning rate. Moving average over 50 iterations. Negative cosine similarity (and thus overshooting) even for very small learning rates. Runs with large learning rates still show overshooting when training is not possible anymore.

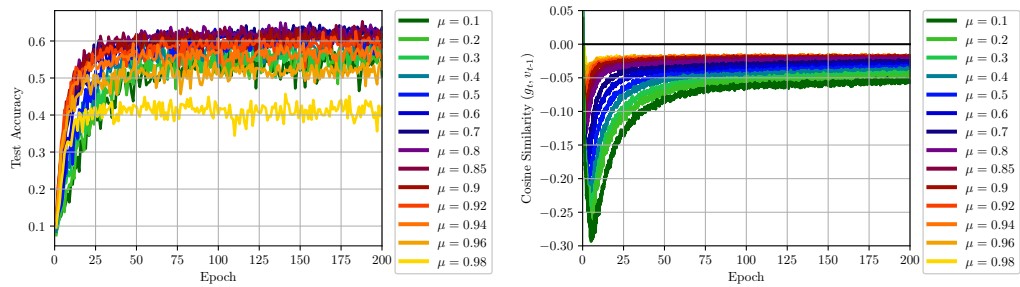

Figure A.6: WRN-16-4 on CIFAR100. Constant learning rate. Moving average over 50 iterations. Small $\mu$, i.e., faster forgetting of the momentum buffer, results in more overshooting.

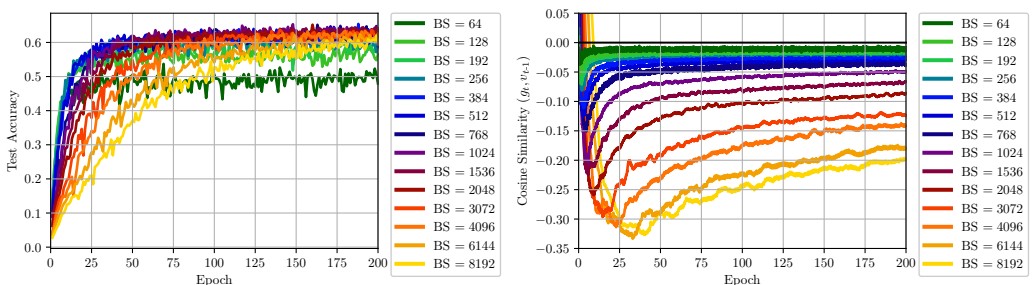

Figure A.7: WRN-16-4 on CIFAR100. Constant learning rate. Moving average over 50 iterations. While larger batch sizes result in less stochasticity, the cosine similarity is reduced. This supports that the negative slope is a property of the training-dataset loss landscape and not caused by limited sampling.

## A.4 Curvature

We calculated the loss curvature in momentum direction, gradient direction and in random direction in Fig. A.8 if training with SGD, SAM and MSAM for WRN-16-4 on CIFAR100. For this we calculate $\epsilon^T \text{Hess}(L_{\mathcal{S}}(w))\epsilon$ for direction vectors $\epsilon$ (normalized to $||\epsilon||_2 = 1$) every 50 optimizer steps.

The curvatures in momentum directions are larger than the curvature random direction (which tends towards the mean curvature as amount of parameters increase) for all optimizers and epochs, validating Setting 1 in Appx. A.1 and thus the suitability of momentum directions for sharpness estimation (especially compared to random perturbations; cf. Appx. A.7).

Additionally, the curvature in these directions offers a measure for the loss sharpness. Since a local minimum of high curvature is approached, all three curvatures increase at the end of the training for SGD. Similarly to Fig. 4, SAM and MSAM are reducing the curvature best in their corresponding perturbation direction and MSAM yields lower curvatures than SAM in random directions.

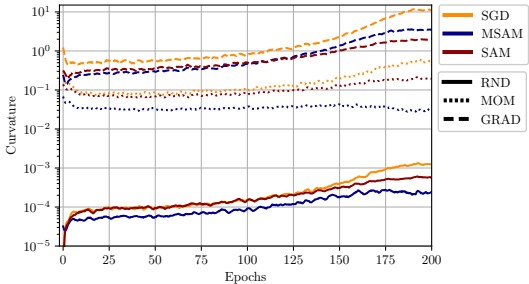

Figure A.8: Curvature of random directions (RND), momentum (MOM) and gradient (GRAD) for different optimizers.

## A.5 Eigenspectrum Analysis

We calculated the top 5 eigenvalues of the Hessian using the Lanczos algorithm for WideResNet-16-4 trained by SGD, SAM and MSAM on CIFAR-100.

Table A.1: Top 5 Hessian eigenvalues.

|      | $\lambda_1$ | $\lambda_2$ | $\lambda_3$ | $\lambda_4$ | $\lambda_5$ |
|------|-------|-------|-------|------|------|
| SGD  | 15.19 | 13.79 | 13.23 | 9.98 | 9.23 |
| MSAM | 4.15  | 3.03  | 2.30  | 1.88 | 1.81 |
| SAM  | 5.18  | 4.85  | 4.43  | 3.72 | 3.36 |

While SAM already effectively reduces the eigenvalues of the Hessian (and thus the curvature), MSAM results in even lower Hessian eigenvalues. These findings align with the other sharpness metrics discussed and further support the conclusion that MSAM leads to lower sharpness than SAM (and SGD).

## A.6 Feature Rank Analysis

Although MSAM approximates the loss calculation on the full training dataset more closely and thus yields lower sharpness (measured as in Fig. 4C) and lower curvature (Fig. A.8, RND direction), we do not see a positive correlation with generalization compared to SAM. Similarly to other authors (Andriushchenko *et al.*, 2023b; Dinh *et al.*, 2017; Springer *et al.*, 2024), we also affirm, that explanations for the improved generalization of SAM/MSAM go beyond direct causal relations to sharpness minimization. In analogy to Mueller *et al.* (2023) and Andriushchenko *et al.* (2023a), we investigated the hypothesis, that the improved generalization of SAM stems from low rank features. For WRN-16-4 trained on CIFAR100 for features after the last convolutional block (before pooling) we found feature ranks of 5019 for SGD, 4775 for SAM and 3791 for MSAM. MSAM reduces the feature rank significantly more than SAM and thus helps to find more distinct and task specific features.

## A.7 Random and Last Gradient Perturbations

Instead of the momentum vector $v_t$ in MSAM, we also tried to use other perturbations $\epsilon$ which are independent of the current gradient and thus do not bring significant computational overhead, namely the last iterations gradients $g_{t-1}$ (cf. Daskalakis *et al.* (2018); Lin *et al.* (2020a)) with positive and negative sign as well as Gaussian random vectors (cf. Wen *et al.* (2018)). For each variation, we tested absolute perturbations (Fig. A.9A)

$$\epsilon^{\text{ABS}} = \rho \frac{\delta}{||\delta||} \tag{A.2}$$

and relative perturbations (Fig. A.9B)

$$\epsilon^{\text{REL}} = \rho \frac{\delta|w|}{||\delta w||}, \tag{A.3}$$

with weights $w$ (multiplied element-wise) and, e.g., $\delta = -v_t$ for MSAM.
MSAM provides the only perturbation reaching SAM-like performance without inducing relevant computational overhead.

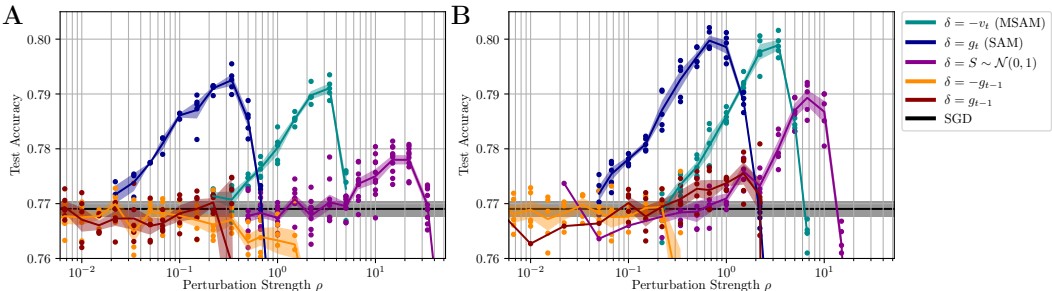

Figure A.9: Random perturbations and last gradient perturbations compared to SAM and MSAM. **A**: Absolute perturbations, **B**: Relative perturbations, i.e., scaled by $|w_i|$ before normalization. All perturbations normalized by L2-norm and scaled by $\rho$. WideResNet 16-4 trained on CIFAR100. MSAM always better than other current gradient-independent perturbations.

## A.8 Hyperparameter Stability

To show the stability of MSAM and its hyper parameter $\rho$, we varied the learning rate $\eta$ and the momentum factor $\mu$ when optimizing a WRN-16-4 on CIFAR100 for fixed $\rho = 2.2$ and depict results in Fig. A.10. MSAM yields stable performance increases compared to SGD and NAG over wide ranges of hyperparameters. We made similar observations when comparing SGD and MSAM for different number of epochs (cf. Fig. A.11).

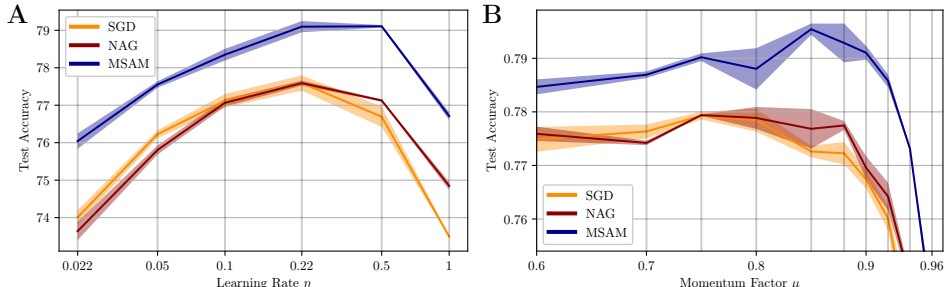

Figure A.10: WRN-16-4 on CIFAR100 fixed $\rho = 2.2$. A: Learning rate $\eta$ ablation (log-scale). B: Momentum factor $\mu$ ablation.

## A.9 Comparison with Same Computational Budgets

We compare MSAM and SAM (and SGD and NAG) when given the same computational budget for WRN-16-4 on CIFAR100 for a wider range of epochs (up to 1200) in Fig. A.11. I.e., running SAM for half the number of epochs compared to other optimizers, resulting in the same number of network passes for all optimizers. MSAM performs similar to NAG (and SGD) for short training times, however, if trained until convergence of SGD/NAG or even longer (overfitting occurs; SGD/NAG results decrease again) MSAM reaches higher test accuracies and overfitting is prevented. Due to the additional forward/backward passes SAM performs worse compared to MSAM for limited computational budgets. For long training times MSAM and SAM do not differ significantly.

We further support these observations by training a ViT-S/32 with MSAM with doubled number of epochs (180) on ImageNet where we reach 70.1% test accuracy and thus clearly outperform SAMs 69.1% (cf. Tab. 2).

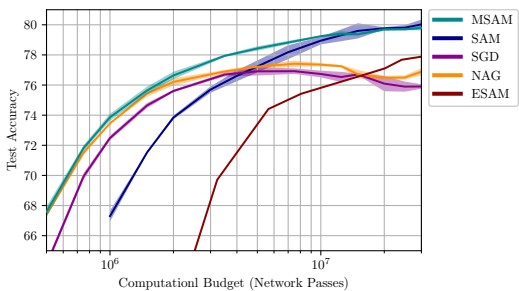

Figure A.11: Comparing different optimizers when given the same computational budget. WRN-16-4 on CIFAR100.

## A.10 Loss Ascent in Momentum Direction

To validate that the perturbation of the loss results in a loss increase, we show the perturbed and unperturbed loss during training for different learning rates in Fig. A.12.

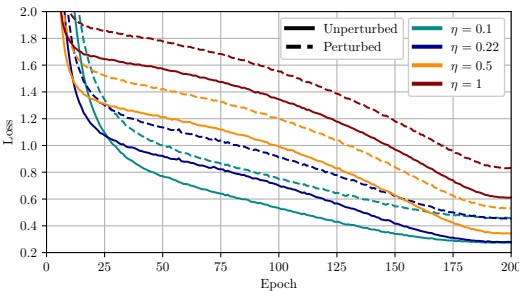

Figure A.12: Loss before $(L_{\mathcal{B}_t}(\boldsymbol{w}_t))$ and after $(L_{\mathcal{B}_t}(\boldsymbol{w}_t - \rho\boldsymbol{v}_t/||\boldsymbol{v}_t||))$ perturbation in momentum direction as done by MSAM ($\rho = 3$) for WRN 16-4 on CIFAR100 for different learning rates.

## A.11 3D Loss Landscapes

We show loss landscapes after training with SGD, SAM and MSAM for two random filter-normalized perturbation directions as done by Li *et al.* (2018) in Fig. A.13. The sharpness/loss landscapes show rotational symmetry. SAM and MSAM both reduce sharpness effectively, while MSAM reaches an even flatter minimum.

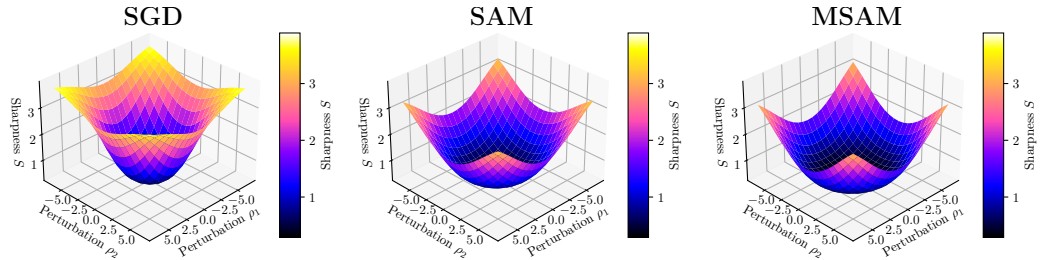

Figure A.13: 2D sharpness landscapes for two random filter-normalized perturbations (cf. Li *et al.* (2018)) as shown in Fig. 4C in 1D.

## A.12 Effects of MSAM/SAM During ViT's Warm-up Phase

We further investigate the effect of SAM/MSAM during warm-up phase in Tab. A.2. As described above, we do not apply MSAM during the warmup phase by default (i.e. setting $\rho = 0$) since if doing so, we observe a drop in test accuracy from 69.1% to 66.1% which is below the AdamW

baseline. We assume fluctuations of the momentum vector, that determines the perturbation direction for MSAM, to cause instabilities during the warmup phase. A similar effect can be seen for SAM, however, it is less pronounced, so that applying perturbations during the warm-up phase does not thwart SAM hugely. Since we focused on proposing a computationally more efficient variant and not on improving the generalization of SAM in this work, we thus decided to stay consistent with related work and conduct our extensive experiments in Tab. 2 while applying SAM also during the warm-up phase. Nevertheless, we would generally propose to apply SAM only after the warmup phase for ViT models to further improve SAM. We think further investigating effects of $\rho$-scheduling for SAM and MSAM is of high interest. E.g., Zhuang *et al.* (2022) investigated to reduce $\rho$ during training (contrary to what our findings suggest) by binding it to the learning rate scheduling for SAM and they did not notice benefits. Despite the discussion in Appx. A.2, we could not observe analogous effects for ResNets (though we did not study these extensively).

Table A.2: Impact of application of SAM/MSAM during warm-up phase. ViT S/32 on ImageNet. By default MSAM is applied after warmup phase only while SAM is always applied.

| AdamW | SAM | SAM (after warmup only) | MSAM | MSAM (during warmup) |
|-------|-----|-------------------------|------|----------------------|
| 67.1  | 69.2 | 69.8                   | 69.1 | 66.1                 |

### A.13 Training and Implementation Details

If not stated differently, we calculate uncertainties of mean accuracies by $68\%$ CI estimation assuming Student's t-distribution.

We tuned weight decay and learning rates for our baseline models (SGD/AdamW) and did not alter these parameters for the other used optimizing strategies. We used basic augmentations (horizontal flipping, cutout and cropping) for CIFAR100 trainings and normalized inputs to mean 0 and standard deviation 1. For ImageNet trainings we used Inception-like preprocessing (Szegedy *et al.*, 2015) with 224x224 resolution, normalized inputs to mean 0 and std 1 and clipped gradients L2-norms to 1.0. We used ViT variants proposed by Beyer *et al.* (2022). A full implementation comprising all models and configuration files is available at https://github.com/MarlonBecker/MSAM. Experiments were performed on up to 4 NVIDIA A100 GPUs.

Table A.3: Training Hyperparameters

|                | CIFAR100 | | ImageNet | |
|----------------|----------|----------|----------|-----|
|                | WideResNets | ResNet50 | ResNets | ViTs |
| Base Optimizer | SGD | SGD | SGD | AdamW |
| Epochs         | 200 | 200 | 100 | 90/300 |
| Learning Rate  | 0.5 | 0.1 | 1 | 1e-3 |
| LR-Scheduler   | cos | cos | cos | cos + linear warm-up (8 epochs) |
| Label Smoothing | 0.1 | 0.1 | 0.1 | - |
| Batch Size     | 256 | 256 | 1024 | 1024 |
| Weight Decay   | 5e-4 | 1e-3 | 1e-4 | 0.1 |

### A.14 Details on Optimizer Comparison

We report experimental details on the results presented in Tab. 1 in this section.

To calculate the speed, we conducted a full optimization on a single GPU for each model and dataset combination, normalized the runtime by SGDs runtime and report the average over all runs per combination.

We trained ViT-S/32 on ImageNet for 90 epochs for all models. Further hyperparameters not specific to SAM variants are reported above in Appx. A.13. Due to limited computational capacities and inline with related work, we did not perform runs for multiple random seeds for ImageNet trainings. Thus, we did not report standard deviations for these runs.

We adapted official implementations of ESAM (Du *et al.*, 2022b) and MESA (Du *et al.*, 2022a) while no official implementation was available for LookSAM (Liu *et al.*, 2022).

For LookSAM, we fixed the trade-off parameter $k = 5$ and conducted a thorough search on the

additional hyper parameter $\alpha$, since the value suggested for ViTs by the original authors ($\alpha = 0.7$) was not suitable for our experiments (also see Fig. A.16). We decided to set $\alpha = 0.1$, while runs for $\alpha > 0.3$ did not yield further performance increases. Full hyperparameter search results are reported in Fig. A.14.

ESAM comprises two hyperparameter ($\gamma$ and $\beta$) that steer the performance/runtime tradeoff which we set to match those of the original paper (i.e. $\gamma = \beta = 0.5$).

For MESA we tuned the regularization factor $\lambda$ instead of the perturbation strength $\rho$.

Please also note the full $\rho$ scan results presented in the next section (Fig. A.15 and Fig. A.16)

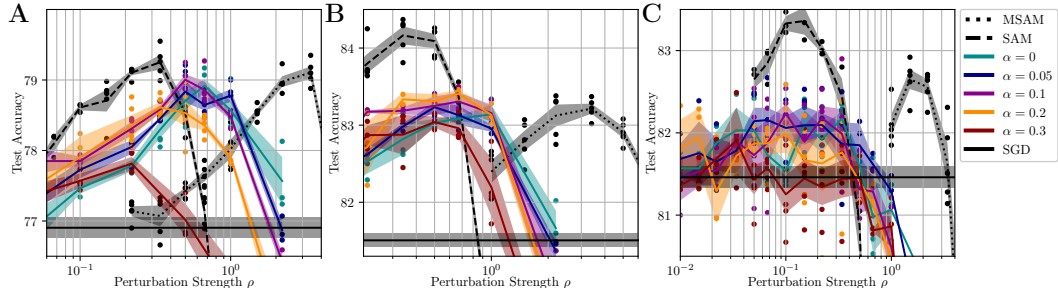

Figure A.14: Full results for $\rho$ and $\alpha$ search for lookSAM ($k = 5$) on CIFAR100. Shaded cone: $68\%$ CI. Dots: Random Seeds. **A**: WideResNet-16-4. **B**: WideResNet-28-10. **C**: ResNet-50.

## A.15    Full Hyper Parameter Search Results

We report our full $\rho$-hyperparameter search results in Fig. A.15, Fig. A.16 and Fig. A.17. In consistency with related work, we report results for best $\rho$ only, in the main text. We sampled $\rho$ with approximately even spacing on logarithmic scale with 6 datapoints per decade, i.e, $\rho \in \{..., 0.1, 0.15, 0.22, 0.34, 0.5, 0.67, 1, 1.5, ...\}$, for experiments on CIFAR100 and with 4 datapoints per decade, i.e, $\rho \in \{..., 0.1, 0.17, 0.3, 0.55, 1, 1.7, ...\}$, on ImageNet for experiments in Sec. 3. We used a slightly denser sampling for the visualizations in Sec. 4, but did not use those results for comparisons against baselines or other methods.

While optimal values for $\rho$ vary slightly between models and datasets, we do not observe higher susceptibility to changes in $\rho$ of MSAM compared to SAM.

Over all models and datasets we find higher optimal $\rho$ values for MSAM compared to SAM. Perturbation vectors are normalized (L2-norm), so we conjecture components for parameters of less importance to be more pronounced for momentum vectors compared to gradients on single batches. For ViT models, we find optimal $\rho$ values to be higher compared to ResNets. If chosen even higher, heavy instabilities occur during training, up to models not converging, limiting performances especially for MSAM. Similar to the observations during warm-up phase discussed above, this effect is more pronounced for MSAM. Notably, MSAM looses most performance against SAM on the biggest ViT models and if trained for 300 epochs, when highest $\rho$ values are optimal for SAM. This suggests, that even better performances might be achievable for MSAM if the instability problems are tackled. Strategies to do so might include e.g. clipping $\epsilon$ or scheduling of $\rho$, which we intend to pursue in future work.

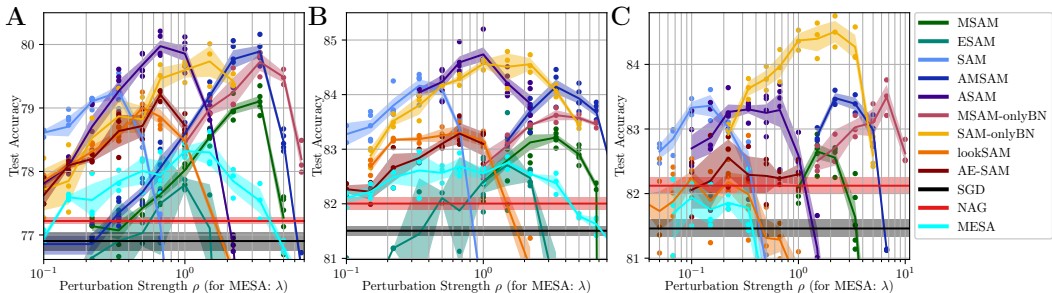

Figure A.15: Full results for $\rho$-search for different SAM/MSAM variants on CIFAR100. AM-SAM/ASAM refer to adaptive-MSAM/adaptive-SAM as in Kwon *et al.* (2021). For LookSAM we set $k = 5$ and report only the best performing value of $\alpha = 0.1$ (cf. Fig. A.14). For MESA: $\lambda$-search results plotted on same axis. Shaded cone: $68\%$ CI. Dots: Random Seeds. **A**: WideResNet-16-4. **B**: WideResNet-28-10. **C**: ResNet-50.

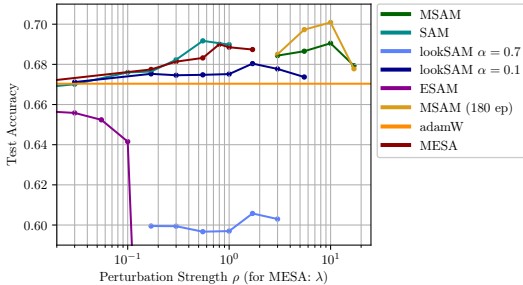

Figure A.16: Full results for $\rho$-search for different SAM/MSAM variants for ViT-S/32 trained for 90 epochs on ImageNet. For MESA: $\lambda$-search results plotted on same axis.

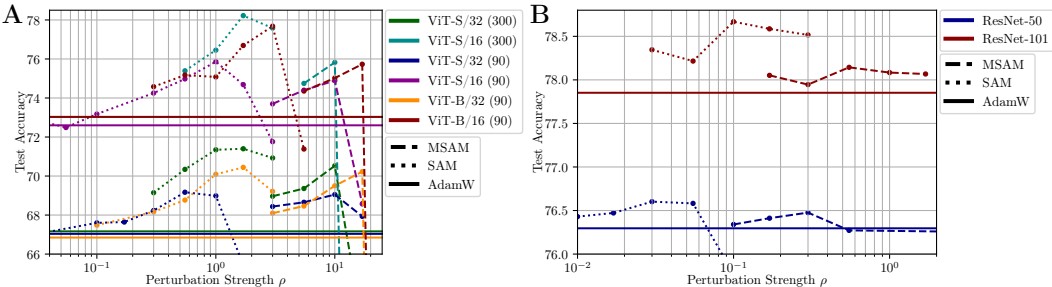

Figure A.17: Full results for $\rho$-search for all models tested on ImageNet (see Tab. 2). **A**: Vision Transformer (epochs in parentheses). **B**: ResNets.

### A.16  Adversarial Robustness

We evaluated the adversarial robustness of MSAM under attacks by PGD (Madry *et al.*, 2018), FGSM Goodfellow *et al.* (2018) and APGDT Croce and Hein (2020). We test several attack methods on a WideResNet-16-4 trained on CIFAR-100. We used default hyperparameters and $\sigma = 0.1$ for gaussian noise and $\epsilon = 1/255$ for the other attack methods. Results shown in A.4 were averaged over 3 randomly initialized runs per optimizer. MSAM improves adversarial robustness similarly to SAM. In particular, under an undirected Gaussian noise attack, MSAM slightly outperforms SAM, further supporting its effectiveness in sharpness minimization. These findings are consistent with those of Zhang *et al.* (2024).

Table A.4: Adversarial Robustness of MSAM compared to SAM and SGD

|  | Gaussian Noise | PGD | FGSM | APGDT |
|---|---|---|---|---|
| SGD | 22.55 | 24.05 | 31.80 | 20.72 |
| MSAM | 26.86 | 28.92 | 35.46 | 27.06 |
| SAM | 25.77 | 30.22 | 35.30 | 26.08 |

