# OpenReview forum: "Momentum-SAM: Sharpness Aware Minimization without Computational Overhead"
_NeurIPS.cc/2025/Conference — NeurIPS 2025 poster_

### Official Review · Reviewer_Uekb · 2025-06-28

**Clarity:** 3
**Significance:** 3
**Originality:** 3
**Rating:** 5
**Confidence:** 2

**Summary:**

This paper introduces Momentum Sharpness-Aware Minimization (MSAM), a novel optimizer designed to reduce the sharpness of training loss surfaces for deep neural networks without incurring the significant computational and memory overhead of SAM. Unlike SAM, which perturbs parameters along the gradient direction and requires an extra gradient computation, MSAM leverages the accumulated momentum vector for perturbation, achieving sharpness-aware updates with virtually no additional cost over standard optimizers like SGD or Adam. The authors provide a detailed empirical evaluation of MSAM on multiple benchmark datasets and architectures, demonstrating that MSAM achieves generalization performance comparable to SAM while being much more computationally efficient. The paper also offers an in-depth analysis comparing MSAM, Nesterov Accelerated Gradient (NAG), and SAM, and provides new insights into the effects of perturbing in the momentum direction, including the impact of negative momentum perturbations on the loss landscape.

**Questions:**

1. Although the appendix contains some theoretical analysis for MSAM, the content and proofs are rather brief, and there does not appear to be a convergence proof. I suggest that key theoretical assumptions and results should be presented in the main text, as they are more convincing than purely empirical conclusions.
2. Since MSAM is easy to implement and incurs computational costs similar to SGD or Adam, why not conduct experiments on different types of tasks, such as natural language processing or large language models, instead of focusing mainly on standard image classification benchmarks? This would help further demonstrate the effectiveness and generalizability of the method across a broader range of application domains.
3. The font sizes in Table 1 compared to Tables 2 and 3 are quite inconsistent, which affects the readability. It would be better to adjust the layout for consistency. Additionally, in Table 2, why are the baselines for ResNet models always SGD and for ViT models always AdamW? Is there a specific reason for this choice?

**Ethical Concerns:**

["NO or VERY MINOR ethics concerns only"]

**Final Justification:**

In the rebuttal, the authors explained the empirical focus of their study and supplemented more experimental results of an NLP task. They have fully addressed my concerns, and thus I raised the score.

**Limitations:**

Yes

**Paper Formatting Concerns:**

No formatting concerns

**Quality:**

3

**Strengths And Weaknesses:**

Strength
1. MSAM maintains the generalization improvements of SAM but removes its major computational bottleneck, making sharpness-aware optimization feasible even when training resources are limited.
2. The algorithm is simple to implement and can be seamlessly integrated into existing training frameworks based on popular optimizers such as SGD or Adam.
3. The authors offer insightful analyses by comparing MSAM, NAG, and SAM, and shed new light on the role of momentum-based perturbations in improving generalization.

Weaknesses
1. The paper lacks rigorous theoretical analysis or convergence guarantees for MSAM, relying instead on empirical results, assumptions, and intuitive explanations.
2. Although the experiments are extensive, the study mostly focuses on standard benchmarks and architectures. The performance of MSAM on larger or more specialized tasks remains unclear.

---

> ### Author Rebuttal · Authors · 2025-07-28
>
> Thank you very much for your positive and constructive feedback.
>
> While we agree that a solid theoretical model to explain the effects of MSAM would be valuable, we would like to emphasize the intentional empirical focus of our study.  Our central finding — a negative correlation between momentum and gradient — is not adequately explained by most existing theoretical frameworks. Instead of relying on heavily simplified assumptions that may hinder the transferability of results to real-world scenarios, we chose to conduct a comprehensive and rigorous empirical investigation of SOTA network architectures and tasks. We will clarify this empirical focus, as well as its limitations, more explicitly in the conclusion section of our manuscript.
>
> Questions:
> 1. Since our method is designed to improve generalization, our theoretical analysis in Section A.1 focuses on a generalization bound rather than a convergence proof, in line with the original SAM paper by Foret et al. (2021). We agree that presenting the main theoretical results in the main text will enhance the clarity and impact of our work, and we will update the manuscript accordingly.
> 2. We conducted additional experiments in the context of language processing. Due to time constraints, we selected a simple task and model: English-to-Romanian machine translation using the WMT 2016 dataset. We fine-tuned a T5-tiny model (efficient variant by Tay et al.) starting from a publicly available checkpoint pretrained on the C4 dataset. We scanned $\rho \in {0.01, 0.03, 0.1, 0.3, 1}$ and found that both MSAM and SAM performed best at $\rho = 0.1$. The resulting BLEU scores were 23.35 for AdamW, 23.57 for SAM, and 23.64 for MSAM. Notably, MSAM slightly outperformed both SAM and AdamW while requiring approximately half the computational cost. We would be happy to include these results to our manuscript.
> 3. Thank you for pointing out the inconsistent font size, we will fix this. Regarding the choice of the baseline optimizers, we used SGD for ResNets and AdamW for ViT, as these give the best performance, and our aim is to compare against the strongest respective baselines.
>
> Please let us know if you have any unresolved concerns about our manuscript.
>
> Tay et al. "Scale Efficiently: Insights from Pretraining and Finetuning Transformers" ICLR 2022

---

> > ### Comment · Reviewer_Uekb · 2025-08-07
> >
> > Thank you for your response and the additional experimental data, which addressed my concerns. I have raised my score.
> > In addition, since the authors emphasize the empirical performance of this work, future experiments on more diverse datasets and models could further highlight the value of this study.

---

> ### Author Response · Authors · 2025-08-06
>
> Thank you again for your review. Following the program chairs' guidelines and the AC's reminder, we would greatly appreciate your thoughts on whether our clarifications and additional experiments sufficiently resolved your concerns. We are very motivated to improve our work and its evaluation and welcome any further feedback.

---

### Official Review · Reviewer_pE5s · 2025-07-01

**Clarity:** 2
**Significance:** 2
**Originality:** 2
**Rating:** 3
**Confidence:** 4

**Summary:**

This paper proposes Momentum-SAM (MSAM), an optimizer that reduces sharpness (improving generalization) without the extra compute cost of SAM. Instead of gradient-based perturbations, MSAM uses the momentum direction to estimate sharpness, avoiding extra forward/backward passes. Experiments on CIFAR100 and ImageNet show MSAM matches or exceeds the performance of faster SAM variants while being nearly as efficient as SGD.

**Questions:**

1. For the comparison with LookSAM, which k did you use? For example, LookSAM-5, LookSAM-10?
2. How is the training speed calculated?
3. How does this approach affect model robustness to adversarial attacks compared to SAM?
4. Are there directions to more formally link MSAM to the original SAM objective in terms of an explicit bound, rather than heuristic similarity?

**Ethical Concerns:**

["NO or VERY MINOR ethics concerns only"]

**Final Justification:**

Thank the authors for the detailed rebuttal. I appreciate the experiments about adversarial robustness of MSAM. In the rebuttal part, there is something not very clear to me about the training time. And I still hope to see some theoretical connection, since for the paper about SAM or its variants, it is usually carefully proved. I will keep my score.

**Limitations:**

limitations regarding stability on large-scale ViT training, potential underperformance in certain configurations, and the heuristic nature of the theoretical connection to SAM are discussed.

**Paper Formatting Concerns:**

None observed—formatting complies with NeurIPS guidelines.

**Quality:**

3

**Strengths And Weaknesses:**

Strengths:
1.  The exposition is clear, including motivating diagrams and an algorithm listing.
2. All hyperparameter scans, code references, and implementation details are provided.

Weaknesses:
1. Although well-discussed in the appendix, the intuition behind perturbing in the negative momentum direction could be more concisely explained in the main text.

2. It would be better to report explicit compute time.

---

> ### Author Rebuttal · Authors · 2025-07-28
>
> Thank you very much for your overall positive and constructive review.
>
> Weaknesses:
> 1. Thank you for considering our detailed appendix. We agree that providing a more concise explanation of the intuition behind perturbing in the negative momentum direction in the main text would improve clarity. We will incorporate it into the main text accordingly.
> 2. We originally reported normalized compute times to simplify comparisons across methods with different configurations, but we agree that providing exact compute times offers a more comprehensive view of our specific experiments. Therefore, we will add additional columns to Table 1 reporting the exact compute times per model. For example, the exact times for ViT-S/32 on ImageNet for 90 epochs are: 11h 55m, SAM: 22h 54m, SGD: 11h 48m.
>
>
>  Questions:
>  1. We set $k=5$ as it was also done by the original authors. We also experimented with $k=10$, which resulted in a faster training speed but worse test accuracy compared to MSAM. For more details about our experiments with other SAM variants, please refer to Section A.14.
>  2. To calculate the training speed, we first average the runtime per optimizer and task across the 5 randomly initialized runs.  We then normalize these times by dividing the baseline optimizer’s runtime (SGD or AdamW, depending on the model) by the runtime of the optimizer under consideration. This normalization results in a speed value of 1.0 for the baseline. Since the variation of these values across different tasks is minimal (typically around $\pm 0.02$), we average them across tasks and report a single speed value per optimizer in Table 1 to improve readability. We will add this detailed description to our concise explanation of the training speed calculation in Section A.14.
>  3. Following your suggestion, we evaluated the adversarial robustness of MSAM. We tested several attack methods on a WideResNet-16-4 trained on CIFAR-100. (We used default hyperparameters and $\sigma=0.1$ for gaussian noise and $\epsilon=1/255$ for the other attack methods. Results were averaged over 3 randomly initialized runs per optimizer.)
> | Optimizer  | gaussian noise  | PGD (Madry et al.)   | FGSM (Goodfellow et al.) |  APGDT (Croce et al.) |
> |---|---|---|---|---|
> | SGD  | 22.55  |  24.05 | 31.80  |  20.72 |
> | MSAM  |  26.86 |  28.92 |  35.46 | 27.06  |
> | SAM  |  25.77 | 30.22  |  35.30 |  26.08 |
>
> MSAM improves adversarial robustness similarly to SAM. In particular, under an undirected Gaussian noise attack, MSAM slightly outperforms SAM, further supporting its effectiveness in sharpness minimization. These findings are consistent with those of Zhang et al. (2024) , which we will add to the related work section.
> Thank you for suggesting this experiment - it aligns very well with our work, and we are pleased to include it in the manuscript.
>
> 4. We formally link MSAM to the original SAM objective by showing in Section A.1 that a generalization bound, similar to that of SAM, can be derived for MSAM. While our main focus is on the empirical study, we appreciate the reviewer's interest in the theoretical connection and will highlight it more in the main text.
>
> Please let us know if you have any unresolved concerns about our manuscript.
>
> Croce et al. "Reliable evaluation of adversarial robustness with an ensemble of diverse parameter-free attacks" ICML 2020\
> Madry et al. "Towards Deep Learning Models Resistant to Adversarial Attacks" ICLR 2018\
> Goodfellow et al. "Explaining and harnessing adversarial examples" ICLR 2014\
> Zhang et al. "On the Duality Between Sharpness-Aware Minimization and Adversarial Training," ICML 2024

---

> > ### Comment · Reviewer_pE5s · 2025-08-08
> >
> > Thank the authors for the detailed rebuttal. I appreciate the experiments about adversarial robustness of MSAM.
> > For "Therefore, we will add additional columns to Table 1 reporting the exact compute times per model. For example, the exact times for ViT-S/32 on ImageNet for 90 epochs are: 11h 55m, SAM: 22h 54m, SGD: 11h 48m." is the 11h55m time corresponding to the proposed method?
> > And I still hope to see some theoretical connection, since for the paper about SAM or its variants, it is usually carefully proved.

---

> > > ### Author Response · Authors · 2025-08-08
> > >
> > > Thank you for your additional feedback.
> > >
> > > Yes, the 11h 55m refers to the runtime for MSAM. Sorry for not making this clear in our initial response.
> > >
> > > In Appendix A.1, we show that the same proof technique used by Foret et al. (2021) for SAM, which is based on a looser bound via random perturbations (Eq. 13 and following in Foret et al., 2021), can also be applied to MSAM. This means that the same theoretical guarantees established for SAM also hold for MSAM.
> > >
> > > Thanks again for your detailed feedback, especially your suggestion to include a robustness analysis. We believe it will be a great addition to our work.

---

### Official Review · Reviewer_Vj9r · 2025-07-03

**Clarity:** 3
**Significance:** 3
**Originality:** 3
**Rating:** 4
**Confidence:** 4

**Summary:**

- This paper aims to improve the efficiency of Sharpness-aware minimization.
- This paper proposes the MSAM, which mimics the framework of Nesterov Accelerated Gradient, using the momentum vector computed in each iteration to approximate the adversarial weight perturbation.
- Empirically, this paper conducts experiments on CIFAR and ImageNet to demonstrate the effectiveness of MSAM (faster than SAM and better than SGD). The paper also provides empirical analysis to show the similarity between gradient directions of MSAM and SAM. Theoretically, the paper provides a loose bound which shows the loss of MSAM upper-bounds the expected loss of general perturbation.

**Questions:**

In terms of performance of MSAM, it highly depends on the perturbation \rho and learning rate. Can you provide more insights on the coupling effect of the two hyper-parameters?  How to ensure the similarity of momentum vector and graident?

**Ethical Concerns:**

["NO or VERY MINOR ethics concerns only"]

**Final Justification:**

Keep score unchanged, boardline accept.

**Limitations:**

Better theory analysis

**Quality:**

3

**Strengths And Weaknesses:**

- The paper is well-written. From method, to experiments, to analysis, they are well-organized
- The observation is counter-intuitive but also interesting. The similarity between momentum vector and gradient is negative. This observation are supported by empirical results. The performance is comparable with SAM and the efficiency is good.

- (Not a weakness at the stage) The theory analysis is not perfect. I encourage more solid analysis on this observation and more extensive experiments (larger model and diverse dataset). If the observation can be further supported by large-scale experiments and more solid theory, It would be perfect.

---

> ### Author Rebuttal · Authors · 2025-07-28
>
> Thank you for your positive and constructive feedback.
>
> While we agree that a solid theoretical model to explain the effects of MSAM would be valuable, we would like to emphasize the intentional empirical focus of our study.  Our central finding — a negative correlation between momentum and gradient — is not adequately explained by most existing theoretical frameworks. Instead of relying on heavily simplified assumptions that may hinder the transferability of results to real-world scenarios, we chose to conduct a comprehensive and rigorous empirical investigation of SOTA network architectures and tasks. We will clarify this empirical focus, as well as its limitations, more explicitly in the conclusion section of our manuscript.
>
> Thank you for highlighting the potential coupling between $\rho$ and the learning rate $\eta$. We examined this coupling, and we found that, as illustrated in Figure A.10, no strong interdependence can be observed.
> A fixed value of $\rho = 2.2$ consistently improves test accuracy across a wide range of learning rates. Additionally, a grid search over $\rho$ and $\eta$ did not reveal any significant dependency between the two parameters (not shown in the current manuscript, but we will add these full results to the appendix).
>
> Additionally,  as shown in Figure A.5, a negative correlation between the momentum buffer and the gradient is consistently observed across the full range of learning rates from $\eta = 0.001$ to $\eta = 2.2$. Figures A.6 and A.7 further illustrate that the similarity between the momentum vector and the gradient is a stable phenomenon, occurring independently of both the momentum coefficient $\mu$ and the batch size.
>
> Please let us know if you have any unresolved concerns about our manuscript.

---

### Official Review · Reviewer_xcYq · 2025-07-08

**Clarity:** 3
**Significance:** 3
**Originality:** 2
**Rating:** 5
**Confidence:** 3

**Summary:**

This paper proposes Momentum-SAM (MSAM), a variant of the Sharpness-Aware Minimization (SAM) optimizer. The main idea is to replace the  gradient ascent step with a perturbation with the momentum vector. The empirical insight is that the momentum vector negatively correlated with the batch gradient. Authors use this as proxy for the gradient ascent step in SAM, thus relacing the need for 2 separate gradient calls. The paper provides empirical evidence for models and datasets to validate the approach.

**Questions:**

1. It would be useful to add an ablation study that isolates the source of the benefit. For example, comparing against a version of MSAM where the SGD step is performed with the raw gradient $g_{MSAM}$ instead of the momentum vector $v_{t+1}$ would clarify whether the performance gain comes purely from the momentum-based perturbation or if the momentum-based weight update is also essential.


2. How does the the eigenspectrum of the Hessian look after MSAM when compared to SGD and SAM? Showing that MSAM, like SAM, suppresses the top eigenvalues of the Hessian would provide stronger evidence that MSAM is proxy for SAM and is exploring flat minimas.

**Ethical Concerns:**

["NO or VERY MINOR ethics concerns only"]

**Final Justification:**

Post rebuttal updates:

- Authors isolated the role of the momentum buffer $v_t$​ by removing its use from the weight update and showed that MSAM retains strong performance even when momentum is excluded from the update, highlighting the benefit of momentum in the perturbation step alone.

- Authors conducted a Hessian analysis, computing the top 5 eigenvalues for SGD, SAM, and MSAM and showed that MSAM achieves the lowest curvature, further confirming its effectiveness.

Authors propose to include these as part of the manuscript and thus I raised my score for the paper.

**Limitations:**

Yes

**Paper Formatting Concerns:**

No.

**Quality:**

3

**Strengths And Weaknesses:**

## Strengths

1. The idea of using the existing momentum vector as a proxy for the gradient ascent step in sharpness-aware perturbation is useful.
2. Overall paper is well written and authors empirically show strong generalization performance of MSAM when compared to baselines like SGD, and comparable performance to SAM and other variants of SAM.


## Weaknesses
- The momentum vector ($v_t$) is used in two distinct places in the proposed algorithm (Algorithm 1):
   * In the "SGD step" to update the weights (w_t+1 = w_t - η * v_t+1).
   * In the "next pert" step to define the perturbation for the next iteration ($w̃_{t+1} = w_{t+1} - ρ * v_{t+1} / ||v_{t+1}||$).
It's not clear where the improved generalization comes from momentum based perturbation, or momentum-based weight is also necessary.

- The momentum vector is negatively correlated with the batch gradient is interesting but an empirical heuristic. I can imagine simple optimization problems where this wouldn't hold. This should perhaps be made clearer in the paper.

---

> ### Author Rebuttal · Authors · 2025-07-28
>
> Thank you for your positive and constructive feedback.
>
> Weaknesses:
> 1. Thank you for highlighting this observation. We hope that our response to your first question resolves your concerns.
> 2. We fully agree that the negative correlation between the momentum vector and the gradient may not hold for many optimization problems beyond deep learning. We will revise the manuscript to emphasize that our proposed algorithm is specifically designed for optimization in the context of deep learning. Additionally, we will clarify that our focus is on  providing a thorough empirical evaluation of SOTA network architectures, rather than on deriving results from a simplified theoretical model.
>
>  Questions:
>  1. Thank you for suggesting the additional ablation experiment. In MSAM, the first effect of the momentum buffer $v_t$ in the weight update step is the same as in standard SGD with momentum. We made no changes in this regard and built MSAM on top of SGD with momentum as well as on top of Adam. As you recommended, we conducted an experiment where we isolated this effect by removing the use of $v_t$ from the weight update step. For the baseline optimizer, this corresponds to training with SGD without momentum (i.e. setting $\mu = 0$). For MSAM we kept $\mu = 0.9$ to compute $v_t$, but used it only in the perturbation step, not in the weight update. For a WideResNet-16-4 trained on CIFAR-100 we obtained the following test accuracies: SGD $75.96 \pm 0.19; MSAM: $77.16 \pm 0.13$. While the performance of SGD drops significantly when momentum is removed, MSAM still delivers substantial improvements. This confirms that the benefit of the momentum-based perturbation in MSAM can be disentangled from the momentum use in the weight update, and still results in significant performance gains.
> 2. We calculated the top 5 eigenvalues of the Hessian using the Lanczos algorithm for WideResNet-16-4 trained by SGD, SAM and MSAM on CIFAR-100.
>
> | Optimizer  | $\lambda_1$  | $\lambda_2$   | $\lambda_3$ |  $\lambda_4$ | $\lambda_5$ |
> |---|---|---|---|---|---|
> | SGD  | 15.19  | 13.79  | 13.23  | 9.98  | 9.23  |
> | MSAM  |  4.15 | 3.03  | 2.30  | 1.88  | 1.81  |
> | SAM  |  5.18 | 4.85  | 4.43  | 3.72  |  3.36 |
>
> While SAM already effectively reduces the eigenvalues of the Hessian (and thus the curvature), MSAM results in even lower Hessian eigenvalues. These findings align with the other sharpness metrics discussed in our manuscript and further support the conclusion that MSAM leads to lower sharpness than SAM (and SGD). We appreciate your suggestion to include this experiment, which we think is a valuable addition to our work, particularly as a complement to the curvature analysis presented in Section A.4, and will incorporate it into the revised manuscript.
>
> Please let us know if you have any unresolved concerns about our manuscript.

---

> > ### Comment · Reviewer_xcYq · 2025-08-06
> > **Thank you for running the experiments!**
> >
> > Thank you for the rebuttal and for running the two experiments I was curious about. I believe including these results in the next version of the paper would help strengthen the work. Additionally, revising the paper to better emphasize the context of deep learning for the proposed optimization methodology would further improve its clarity.

---

> ### Author Response · Authors · 2025-08-06
>
> Thank you again for your review. Following the program chairs' guidelines and the AC's reminder, we would greatly appreciate your thoughts on whether our clarifications and additional experiments sufficiently resolved your concerns. We are very motivated to improve our work and its evaluation and welcome any further feedback.

---

### Comment · Area_Chair_qzA9 · 2025-08-05

Dear Reviewers, This is a gentle reminder to read the author's rebuttal and engage in the discussion. The discussion period ends on August 6th AoE. Thank you.

---

> ### Comment · Area_Chair_qzA9 · 2025-08-05
>
> Dear Reviewers, the author-reviewer discussion period has been extended to August 8th AoE. Please use this extra time to read the author's rebuttal and engage in discussion with the authors. Thank you.

---

### Note · Authors · 2025-08-11

We believe that the additional experiments, particularly the eigenspectrum analysis and the evaluation of robustness to adversarial attacks, have enhanced the depth and quality of our evaluation. The new findings provide further evidence for our proposed efficient sharpness reduction method.
Furthermore, the additional ablation studies suggested by the reviewers have confirmed our initial claims, thereby increasing the validity of the manuscript.
We agree with the reviewers' critique that the scope of our work should be stated more clearly, namely that it is a detailed empirical study focused on deep learning. We will also emphasize the connection to the theoretical background of SAM more explicitly.
We are confident that we have addressed all questions raised by the reviewers, leaving no unresolved concerns during the discussion period.
In conclusion, we thank the reviewers for their constructive feedback, which has significantly strengthened our work.

---

### Decision · Program_Chairs · 2025-09-17

**Decision:**

Accept (poster)

**Comment:**

This paper introduces Momentum-SAM (MSAM), an optimizer that addresses the computational overhead of Sharpness-Aware Minimization (SAM). The core contribution is the use of the existing momentum vector for parameter perturbation, which avoids the need for a second gradient computation. The authors provide a comprehensive empirical study demonstrating that MSAM achieves generalization performance comparable to SAM with significantly better efficiency.

The primary strength of this work lies in its practicality and efficiency. MSAM offers a powerful solution to a major bottleneck in SAM, making sharpness-aware optimization more accessible. The method is built on a compelling empirical observation—the negative correlation between momentum and gradient, which is well-supported by extensive experiments.

The main weakness noted by reviewers is the lack of a deep theoretical analysis or convergence guarantees. The paper primarily relies on empirical evidence and intuitive explanations. While a generalization bound is briefly mentioned, the theoretical connection to the original SAM objective is not as rigorously proven as in some prior works.

The decision to Accept this paper is based on its significant and timely empirical contribution. The proposed method is highly effective, well-validated, and addresses a crucial, real-world limitation of SAM. The core idea is simple and impactful, giving the method potential for wide adoption. This paper presents a solid contribution to the field of optimization and is well-suited for a poster presentation.

During the rebuttal period, reviewers raised key questions about the source of MSAM's benefits (Reviewer xcYq), its theoretical foundation (Reviewer pE5s), and its generalizability to other tasks (Reviewer Uekb). The authors provided an excellent rebuttal. They conducted additional experiments on Hessian analysis and adversarial robustness, which satisfied Reviewer xcYq. They also provided new results for a language processing task, which convinced Reviewer Uekb. While Reviewer pE5s remained unconvinced on the theoretical front and maintained their borderline reject score, the authors' proactive engagement and the quality of their additional experiments were pivotal in the final decision to recommend acceptance.